# PTEN is a protein phosphatase that targets active PTK6 and inhibits PTK6 oncogenic signaling in prostate cancer

Darren J. Wozniak[1], Andre Kajdacsy-Balla[2], Virgilia Macias[2], Susan Ball-Kell[3], Morgan L. Zenner[1], Wenjun Bie[1] & Angela L. Tyner[1]

PTEN activity is often lost in prostate cancer. We show that the tyrosine kinase PTK6 (BRK) is a PTEN substrate. Phosphorylation of PTK6 tyrosine 342 (PY342) promotes activation, while phosphorylation of tyrosine 447 (PY447) regulates auto-inhibition. Introduction of PTEN into a PTEN null prostate cancer cell line leads to dephosphorylation of PY342 but not PY447 and PTK6 inhibition. Conversely, PTEN knockdown promotes PTK6 activation in PTEN positive cells. Using a variety of PTEN mutant constructs, we show that protein phosphatase activity of PTEN targets PTK6, with efficiency similar to PTP1B, a phosphatase that directly dephosphorylates PTK6 Y342. Conditional disruption of *Pten* in the mouse prostate leads to tumorigenesis and increased phosphorylation of PTK6 Y342, and disruption of *Ptk6* impairs tumorigenesis. In human prostate tumor tissue microarrays, loss of PTEN correlates with increased PTK6 PY342 and poor outcome. These data suggest PTK6 activation promotes invasive prostate cancer induced by *PTEN* loss.

[1] Departments of Biochemistry and Molecular Genetics, Chicago, 60607 IL, USA. [2] Pathology, University of Illinois, Chicago, 60612 IL, USA. [3] Global Path Imaging and Consulting, Ltd., Germantown Hills, 61548 IL, USA. Correspondence and requests for materials should be addressed to A.L.T. (email: atyner@uic.edu)

P rostate cancer is the most commonly diagnosed cancer among men with an estimated 161,360 new cases that will be diagnosed in the United States in 2017[1]. Although a majority of prostate cancers are detected early and the 5-year relative survival approaches 100%, five year survival for men with distant metastases is only 29%. Genetic contributions to prostate cancer have been well documented, including loss of tumor suppressors such as PTEN and p53, and activation of oncogenes like SRC, MET, and FGFR but many of the molecular drivers of aggressive prostate cancer remain elusive[2]. Identification of novel proteins that promote prostate tumorigenesis is critical for the development of new therapeutics.

Protein tyrosine kinase 6 (PTK6, also called BRK and Sik) is a non-receptor tyrosine kinase that belongs to a small family of intracellular kinases, which includes FRK and SRMS, and is distinct from the SRC family[3]. PTK6 is primarily expressed in epithelial tissues, with highest levels in the linings of the gastro-intestinal tract[4]. Mice with systemic disruption of *Ptk6* exhibited enhanced growth and delayed differentiation in the small intestine, but they did not have a cancer phenotype[5]. Contrary to initial expectations, *Ptk6*$^{-/-}$ mice were resistant to development of colon[6], breast[7], and skin[8] cancers in different mouse models, highlighting distinct oncogenic roles for the kinase and its context specific functions in vivo.

While PTK6 contains SH3, SH2, and tyrosine kinase domains, it lacks an N-terminal SH4 domain that promotes palmitoylation/myristoylation and membrane association, and a distinct nuclear localization signal. Nevertheless, membrane-associated and nuclear PTK6 have been detected, and have distinct functions[9]. PTK6 has been shown to have both kinase-dependent and

**Fig. 1** Expression of PTEN reduces PTK6 activation in PC3 cells. **a** Total cell lysates were prepared from PC3 stably expressing empty vector, PTEN, or PTEN G129R. Lysates were subjected to immunoblotting and changes in active PTK6 (PY342) and AKT (PS473) were monitored. **b** PC3 cells stably expressing empty vector, PTEN, or G129R were fractionated into membrane, cytoplasmic, and nuclear compartments. Immunoblot analysis was performed with active PTK6 (PY342 PTK6) and total PTK6. PTEN activity was monitored by active AKT signal (PS473 AKT) compared to total AKT. SP1 was used as a nuclear marker control. Activated PTK6 at the membrane was quantified ($n = 3$). **c** Empty vector, wild-type (WT), kinase-dead (KM), or constitutively active (YF) PTK6 constructs were transiently expressed in PC3 cells stably expressing empty vector or PTEN. Total cell lysates were prepared and subjected to immunoblotting. Tyrosine phosphorylation (PY) of proteins was assayed with a mixture of anti-phospho-tyrosine antibodies, PY20 and 4G10. Activation of PTK6 substrates was measured with phospho-specific antibodies. **d** The relative effect of PTEN expression on activation of PTK6, phospho-FAK, and phospho-BCAR1 was quantified ($n = 3$, **$P < 0.01$, *$P < 0.05$). **e** MYC-tagged empty vector (Vec), wild-type (WT), kinase-dead (KM) or constitutively active (YF) PTK6 recombinant mutants were immunoprecipitated from PC3 cells stably expressing empty vector or PTEN. Bound protein was measured by immunoblot. **f** PTEN was immunoprecipitated from PC3 cells stably expressing empty vector, PTEN, or PTEN G129R. Complexed proteins were assessed by immunoblot. Arrowhead denotes protein of interest. The heavy band at 55 kDa is background and represents IgG

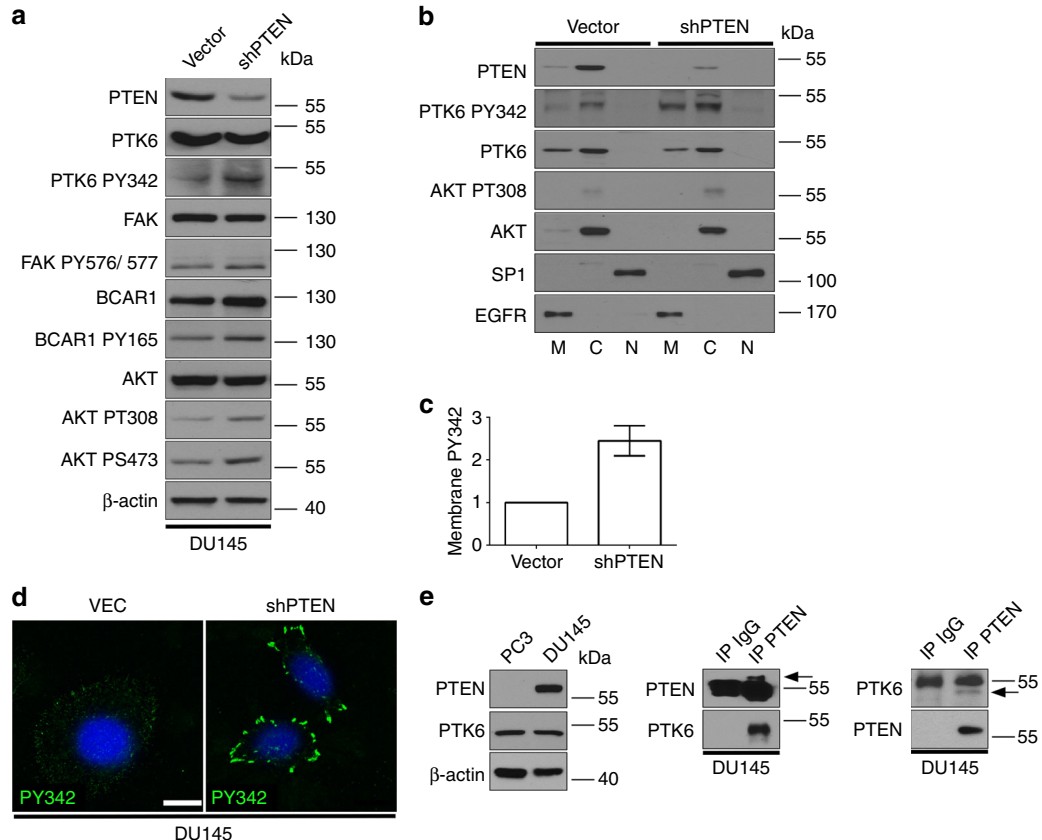

**Fig. 2** Knockdown of PTEN in DU145 cells activates PTK6. **a** DU145 prostate cancer cells were subjected to stable PTEN knockdown by shRNA. Total cell lysates were prepared from DU145 cells with empty Vector or the PTEN shRNA construct (shPTEN). Activation of PTK6 and BCAR1 and FAK, direct targets of PTK6, were monitored by immunoblotting. **b** DU145 cells expressing empty vector or the shRNA targeting PTEN were fractionated into membrane, cytoplasmic, and nuclear compartments and activation of PTK6 was assessed by immunoblotting. SP1 and EGFR were examined as nuclear and membrane markers, respectively. **c** Relative changes in activation of PTK6 at the plasma membrane were quantified by ImageJ ($n = 3$). **d** DU145 cells stably expressing the shRNA targeting PTEN were examined for activation of PTK6 (PY342, green) by immunofluorescence. Cells were counterstained with DAPI (blue). Scale bar, 10 μm. **e** PTEN status in PC3 and DU145 cells was confirmed by immunoblot. PTEN and PTK6 were immunoprecipitated with specific antibodies or non-specific IgG from the same species. Arrowheads point to immunoprecipitated protein. Dark background signal is IgG

independent activities that rely on its enzymatic and adapter protein functions, respectively[10].

In the normal human prostate gland, PTK6 was detected in nuclei of epithelial cells[11]. In the nucleus, PTK6 can inhibit proliferation by negatively regulating activity of the RNA-binding protein Sam68 and inhibiting β-catenin transcription[12, 13]. Nuclear localization of PTK6 is lost in prostate tumors. In the cytoplasm and at the plasma membrane, PTK6 engages oncogenic targets like Paxillin, AKT, BCAR1, and FAK[14]. Membrane-targeted active PTK6 is sufficient to induce transformation of mouse embryonic fibroblasts independent of major tyrosine kinases SRC, YES, and FYN[15]. Activation of PTK6 at the plasma membrane also promotes the epithelial mesenchymal transition (EMT) and survival and metastasis of prostate tumor cells in xenograft models[14]. Activation of PTK6 at the plasma membrane has been observed in human prostate[14], breast[16] and skin tumors[8].

PTEN is a tumor suppressor protein widely recognized as a negative regulator of the PI3K/AKT signaling pathway (reviewed in ref. [17]). PTEN function is frequently lost in prostate cancer and a recent study demonstrated aberrations in the *PTEN* gene in 41% of 150 individuals with metastatic, castration-resistant prostate cancer in a large multiinstitutional sequencing study[18]. Although PTEN acts as a lipid phosphatase that dephosphorylates phosphatidylinositol (3,4,5)-trisphosphate (PIP3) to phosphatidylinositol (4,5)-bisphosphate (PIP2)[19, 20], it was initially identified as a

protein phosphatase that is able to target serine, threonine, and tyrosine residues[21, 22]. More recently, additional phosphatase and PI3K/AKT-independent functions have also been identified for PTEN[23].

Activating phosphorylation of PTK6 at tyrosine residue 342 is increased in cells lacking functional PTEN. The active pool of PTK6, phosphorylated on tyrosine residue 342, is localized to the plasma membrane in the PTEN-null prostate cancer cell line PC3, and in mouse prostates with conditional *Pten* deletion in the prostate[14]. These observations led us to hypothesize that PTEN negatively regulates activation of PTK6. Here, we identify PTK6 as a novel protein substrate of PTEN, and show PTEN is able to dephosphorylate PTK6 specifically on tyrosine residue 342 and attenuate its kinase activity. Reintroduction of PTEN into prostate cancer cells lacking functional PTEN leads to PTK6 dephosphorylation and inhibition of its activity and downstream oncogenic signaling. PTEN directly interacts with PTK6 through association with its catalytic domain. Using a transgenic mouse model, we found that *Ptk6* null mice are more resistant to developing invasive prostate adenocarcinoma following prostate-specific *Pten* deletion. Examination of PTK6 and PTEN in a human prostate tumor tissue microarray demonstrates an inverse correlation between PTK6 activation and expression of PTEN. Inhibition of PTK6 signaling in prostate cancers with loss of PTEN could provide therapeutic benefits and merits further investigation.

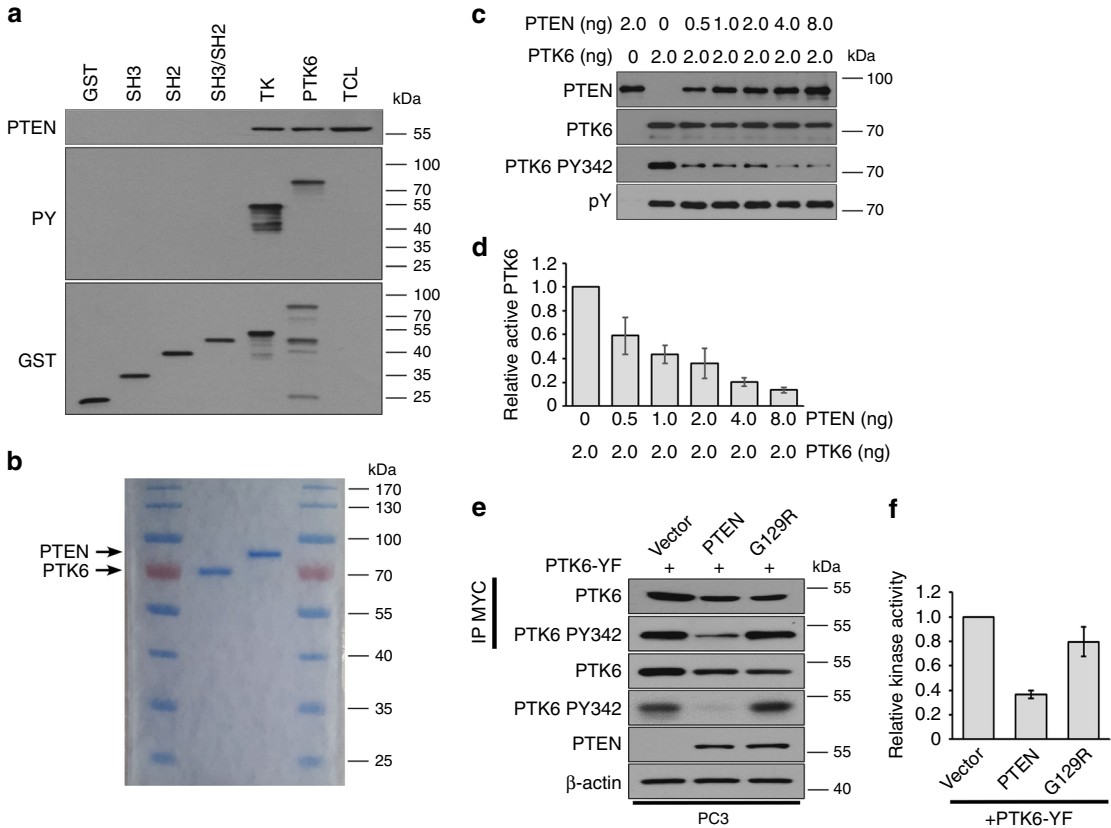

**Fig. 3** PTEN inhibits PTK6 kinase activity. **a** Glutathione beads were used to pull down GST-tagged PTK6 domains incubated with DU145 cell lysates. Bound PTEN protein was observed by immunoblot. **b** Full-length human PTEN and full-length human PTK6 were expressed in *E. coli* and purified using glutathione beads. Eluted protein was subjected to SDS–PAGE and visualized by Coomassie staining to evaluate purity. **c** Purified GST-tagged PTEN and PTK6 were used in an in vitro phosphatase assay. Residual activating phosphorylation of PTK6 was monitored by immunoblotting. **d** Quantification of in vitro phosphatase assays. Data represent three independent experiments run in triplicate. Error bars denote standard error. **e** MYC-tagged constitutively-active PTK6 (PTK6-YF) was immunoprecipitated from PC3 cell expressing empty vector, PTEN, or PTEN G129R. **f** Immunoprecipitated PTK6-YF was used in an in vitro kinase assay. Kinase activity was recorded using an anti-phospho-tyrosine antibody linked to horseradish peroxidase (HRP) in the presence of a light-emitting substrate by measuring the absorbance at 450 nm (*n* = 3 independent trials run in triplicate)

## Results

### PTEN inhibits PTK6 activity and downstream signaling in prostate cancer cells.

Earlier we reported that conditional disruption of *Pten* in the mouse prostate led to activating phosphorylation of membrane-associated PTK6[14] supporting our hypothesis that PTEN may directly negatively regulate PTK6 activity and/or membrane localization in human prostate cancer cells. Wild type HA-tagged PTEN and the dual phosphatase-deficient PTEN G129R mutant[24] were expressed in the *PTEN*-null prostate cancer cell line PC3 that expresses active membrane-associated PTK6 phosphorylated on tyrosine residue 342 (PY342). Cells transfected with empty vector alone retain PY342 (Fig. 1a). When functional PTEN is reintroduced into these cells, levels of active PTK6 PY342 are reduced. The reduction in active PTK6 is most apparent at the plasma membrane where PY342 is enriched (Supplementary Fig. 1a). The phosphatase activity of PTEN is required for the inhibition of active PTK6, as phosphatase-deficient PTEN-G129R has no effect on PTK6 activation status. When these cell lines are fractionated into membrane, cytoplasmic, and nuclear portions, cells with empty vector display active PTK6 PY342 at the plasma membrane and in the cytoplasm. Reduced activation of PTK6 at the plasma membrane and in the cytoplasm is observed in the presence of wild type PTEN, but not PTEN G129R (Fig. 1b). Phosphorylation of AKT on serine residue 473 was examined as a control for

PTEN activity. These data demonstrate that activation of PTK6 is negatively regulated by PTEN and dependent on the phosphatase activity of PTEN.

Since PC3 cells express relatively low levels of PTK6, we transiently expressed MYC-tagged wild-type (WT), kinase-dead (K219M, denoted KM) or inhibition-defective (Y447F, denoted YF) recombinant mutants of PTK6 in stable cell lines containing empty vector (VEC) or the wild-type PTEN expression construct (PTEN) to examine the impact of PTEN on PTK6 downstream signaling. Transiently transfected PTK6-YF is highly phosphorylated at tyrosine residue 342 in PC3 cells, and the highest levels of overall protein tyrosine phosphorylation (PY) are observed when constitutively active PTK6-YF is expressed. Both PTK6-WT and PTK6-YF induce wide spread tyrosine phosphorylation of proteins showing that PTK6 activates several signaling cascades. In the presence of wild type PTEN, PTK6 PY342 but not PY447 is reduced, and activation of downstream signaling is decreased (Fig. 1c).

To assess the effect of PTEN on specific targets of PTK6 signaling, we checked phosphorylation of PTK6 substrates focal adhesion kinase (FAK)[15] and breast cancer anti-estrogen resistance protein (BCAR1, also called p130CAS)[25]. PTK6 phosphorylates FAK and protects cells from anoikis through AKT. PTK6 induces formation of peripheral adhesion complexes and cell migration by directly phosphorylating

BCAR1. In the PC3 VEC cells, PTK6-YF overexpression induces phosphorylation of FAK at tyrosine residues 576/577 and BCAR1 at tyrosine residue 165. When PTEN is ectopically expressed, PTK6-mediated activation of FAK and phosphorylation of BCAR1 is diminished to levels comparable to the empty vector control (Fig. 1c, d). These data demonstrate that PTEN negatively regulates PTK6 signaling and PTK6-mediated phosphorylation of oncogenic substrates FAK and BCAR1.

To examine the association between stably expressed wild type HA-PTEN and transiently expressed MYC-tagged PTK6-WT, -KM, and -YF in PC3 cells, we performed coimmunoprecipitation experiments. Immunoprecipitation of MYC-tagged PTK6

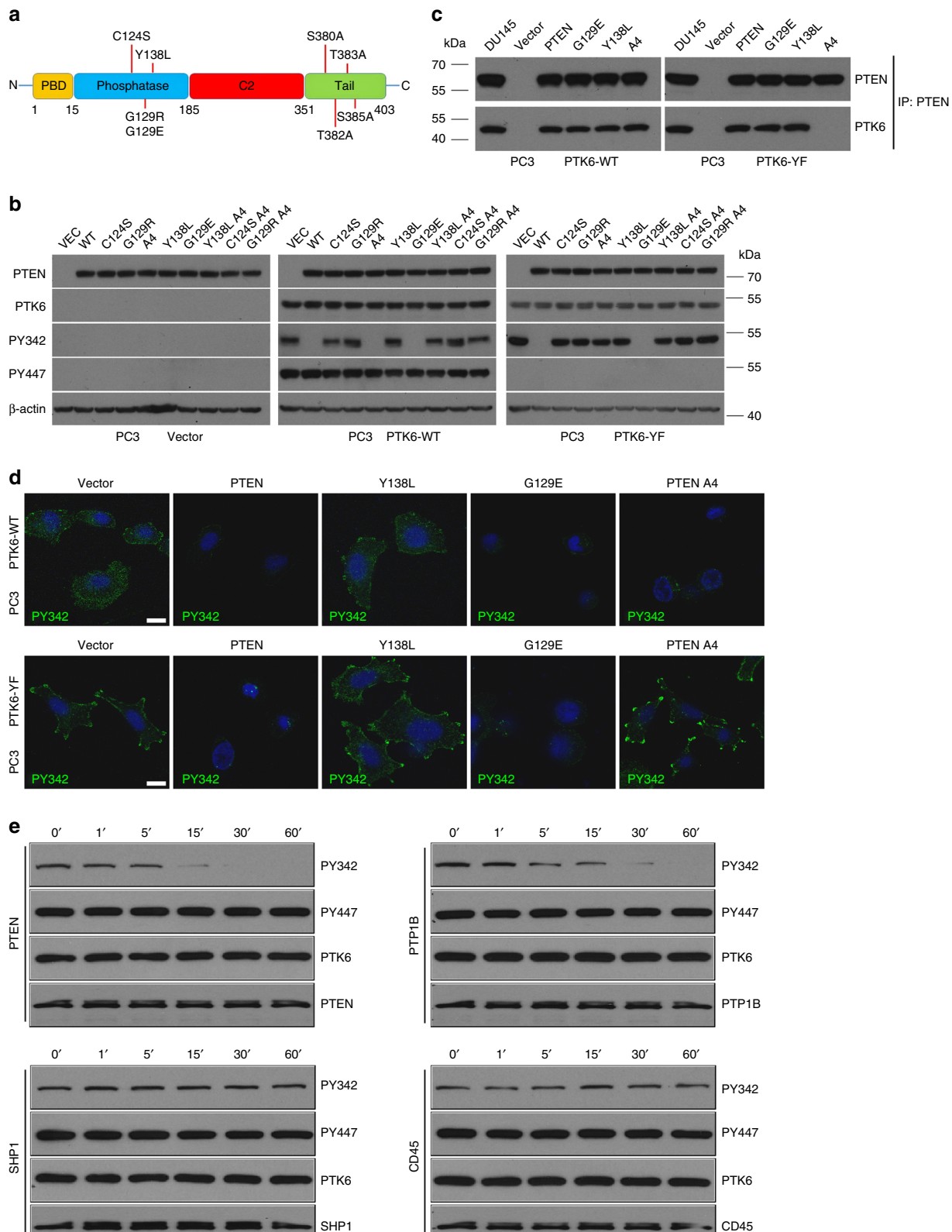

brought down PTEN, while immunoprecipitation of HA-PTEN brought down PTK6 (Fig. 1e). We were also able to coimmunoprecipitate endogenous PTK6 with PTEN from PC3 cells expressing ectopic wild type PTEN, but not phosphatase-defective PTEN G129R, suggesting PTEN catalytic activity may play a role in PTEN-PTK6 association or the bulky positively charged arginine residue in the PTEN C129R mutant disrupts the complex between PTK6 and PTEN (Fig. 1f).

**Knockdown of PTEN induces activation of PTK6.** The prostate cancer cell line DU145 expresses functional PTEN and PTK6 (Fig. 2a). To assess whether PTEN negatively regulates PTK6 activation and signaling, we subjected DU145 cells to stable PTEN knockdown by shRNA, and used selected pools of stable knockdown cells at early passages. Upon PTEN knockdown, we observed increases in active AKT as expected and PTK6 PY342. Additionally, loss of PTEN expression yielded increases in tyrosine phosphorylation of PTK6 substrates FAK and BCAR1 (Fig. 2a). These data show that knockdown of PTEN in prostate cancer cells is sufficient to activate PTK6 and promote its downstream oncogenic signaling.

The stable DU145 cell lines were fractionated into membrane, cytoplasmic and nuclear fractions and localization of EGFR, AKT, and SP1 were examined as controls (Fig. 2b). Upon PTEN knockdown, activation of PTK6 is observed most strikingly at the plasma membrane (Fig. 2b–d) where it has been shown to induce the EMT, at least in part through AKT[9, 14]. We examined localization of active PTK6 in vector control and shPTEN DU145 cells by immunofluorescence. Low levels of active PTK6 PY342 are diffusely distributed throughout the cytoplasm with little enrichment at the plasma membrane in control cells (Fig. 2d). Stable knockdown of PTEN induces strong activation of PTK6 particularly at the plasma membrane. The accumulation of endogenous active PTK6 at the plasma membrane is similar to peripheral adhesion complexes induced by overexpression of membrane-targeted constitutively-active PTK6 palmitoylated/myristoylated PTK6-Y447F[25].

As PTEN negatively regulated PTK6 phosphorylation at tyrosine residue 342, we examined whether endogenous PTK6 and PTEN form a complex within DU145 prostate cancer cells. PTEN is expressed in DU145 cells but absent in PC3 cells, and these cell lines express comparable levels of PTK6 (Fig. 2e). Immunoprecipitation of PTEN from DU145 cells resulted in coimmunoprecipitation of PTK6 (Fig. 2e middle panel). Immunoprecipitated PTEN is visible above the IgG band (arrow). We also immunoprecipitated PTK6 (arrow, below the IgG band, Fig. 2e right panel), and detected PTEN. Immunoprecipitation with non-specific mouse IgG control antibodies did not pull down either protein. These coimmunoprecipitation experiments demonstrate that endogenous PTK6 and PTEN form a complex in prostate cancer cells.

**PTEN associates with the PTK6 tyrosine kinase domain and inhibits its tyrosine kinase activity.** PTK6 contains SH3 and SH2 domains, which mediate its interactions with substrates and other proteins. The interaction between PTK6 and its targets FAK and BCAR1 requires the SH2 domain, while the interaction between PTK6 and AKT mainly involves the SH3 domain[15, 26]. We performed pull-down assays with purified GST-PTK6 domain fusion proteins to identify domains of PTK6 that interact with endogenous PTEN. Interestingly, neither the SH3 nor SH2 domains of PTK6 mediate interaction with PTEN. Only the tyrosine kinase (TK) domain and full-length PTK6 were able to pull-down PTEN from DU145 cell lysates (Fig. 3a). This suggests that PTEN interacts with PTK6 through its tyrosine kinase catalytic domain.

Given that PTK6 interacts with PTEN through its kinase domain, we tested whether PTEN directly dephosphorylates PTK6 at tyrosine residue 342. Recombinant GST-tagged PTK6 and PTEN proteins were overexpressed and purified from *E. coli*. PTK6 is able to autophosphorylate itself at Y342[27] and Y447[28] in bacteria. A Coomassie stained gel showing the purity of the proteins is shown in Fig. 3b. Increasing amounts of PTEN were added to a constant amount of active PTK6. Using an in vitro phosphatase assay, we observed that PTEN was able to dephosphorylate PTK6 at tyrosine residue 342 in a dose-dependent manner. (Fig. 3c, d). These data show that PTEN dephosphorylates PTK6 at Y342 in vitro.

As we use phosphorylation of Y342 as a marker for PTK6 activation, we wanted to confirm that reduced tyrosine phosphorylation at Y342 resulted in diminished tyrosine kinase activity. To do this, we utilized a commercially available universal tyrosine kinase assay kit. PTK6-YF was transiently transfected and immunoprecipitated from the PC3 Vector, PTEN, and PTEN-G129R stable cell lines (Fig. 3e). PTK6-YF was highly phosphorylated at Y342 in the PC3 Vector control cell line, but this activation mark was lost in the PC3 cells stably expressing wild type PTEN (Fig. 3e). Moreover, the enzymatic activity is required for this effect as PTK6-YF activation is not affected in the PC3 cells stably expressing PTEN G129R. Immunoprecipitated protein was assayed for intrinsic kinase activity in the presence of phosphatase inhibitors. PTK6-YF protein from PC3 PTEN cells displays lower kinase activity than PTK6-YF from PC3 Vector and PC3 PTEN-G129R cells (Fig. 3f). These observations support our hypothesis that PTEN negatively regulates PTK6 tyrosine kinase activity by targeting PY342.

**The protein phosphatase activity of PTEN is required for PTK6 inhibition.** To confirm that PTEN is a protein phosphatase that targets PTK6 activating phosphorylation, PC3 cells stably expressing membrane targeted palmitoylated/myristoylated (Palm), wild type (WT), or active (YF) PTK6[29] were transiently transfected with a variety of PTEN expression constructs with mutations diagrammed in Fig. 4a. These include constructs encoding dual phosphatase-deficient PTEN C124S[19] and PTEN

---

**Fig. 4** PTEN protein phosphatase activity is sufficient to inhibit activation of PTK6. **a** A graphical representation of the PTEN domain structure. Mutations in the phosphatase and tail domains including C124S (phosphatase-dead), G129R (phosphatase-deficient), A4 (tail mutant), Y138L (protein phosphatase-dead), G129E (lipid-phosphatase dead), Y138L A4 (protein phosphatase-dead, tail mutant), C124S A4 (phosphatase-dead, tail mutant), or G129R A4 (phosphatase-deficient, tail mutant) are shown. **b** PC3 cells stably expressing empty vector, Palm-WT, or Palm-YF were transiently transfected with empty vector, wild-type PTEN, or recombinant PTEN mutants. Changes in phosphorylation on PTK6 Y342 and Y447 were monitored by immunoblotting. **c** PTEN was immunoprecipitated from PC3 cells stably expressing PTK6-WT and constitutively-active PTK6-YF which were transiently transfected with empty vector, wild-type PTEN, or PTEN mutants G129E, Y138L, or PTEN A4. Bound PTK6 protein was subjected to SDS–PAGE and observed by western blot **d** PC3 cells stably expressing PTK6-WT and constitutively-active PTK6-YF which were transiently transfected with empty vector, wild-type PTEN, or PTEN mutants G129E, Y138L, or PTEN A4. Cells were then probed for PTK6 PY342 (green). Cells were counterstained with DAPI (blue). Scale bar, 10 μm. **e** Purified PTEN, PTP1B, SHP1, and the cytoplasmic catalytic domain of CD45 proteins were used in a phosphatase assay using PTK6 as a substrate. The reaction was run for the described time interval. Changes in PTK6 phosphorylation at Y342 and Y447 were detected by immunoblotting

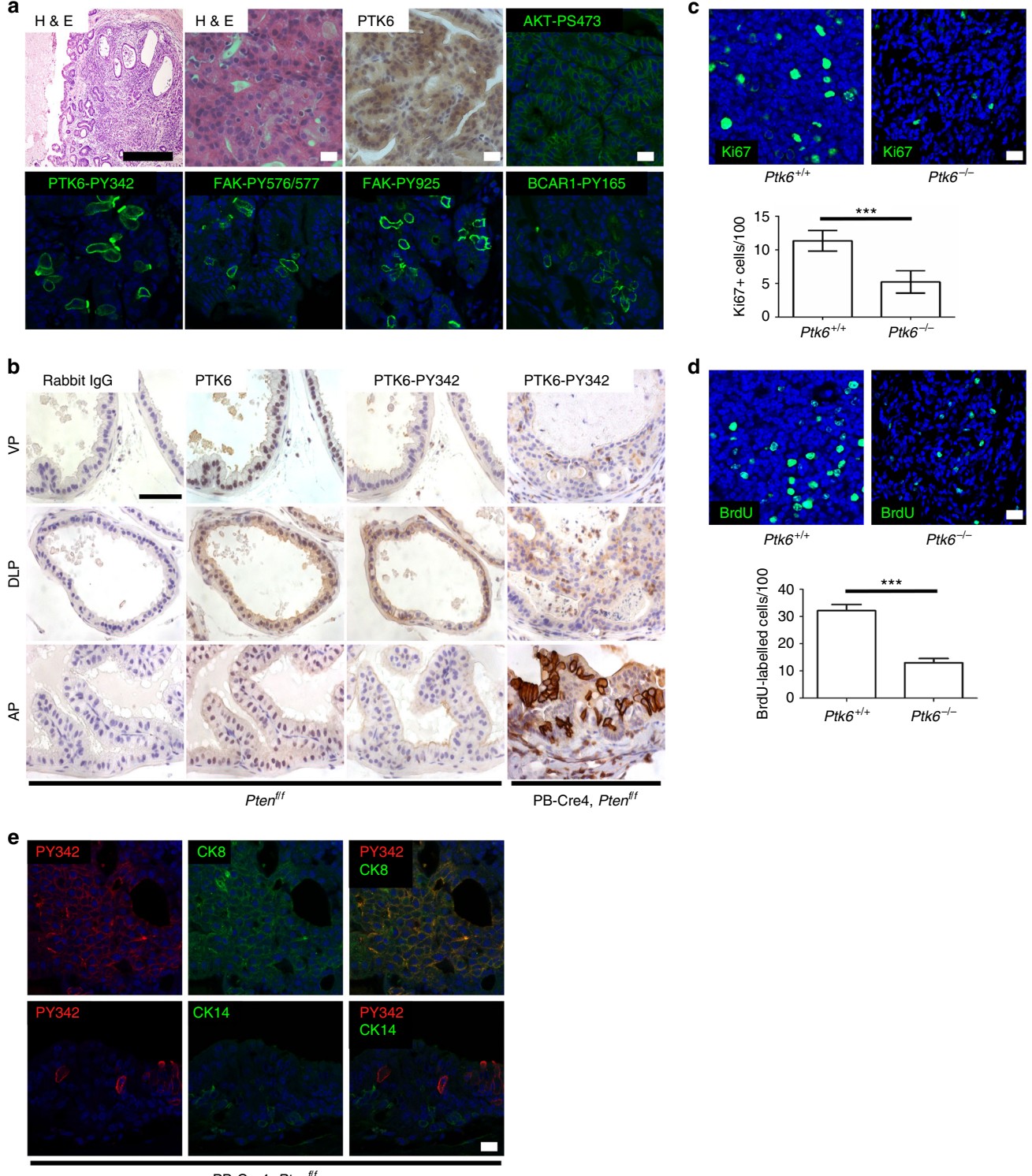

**Fig. 5** PTK6 promotes tumorigenesis following PTEN loss of function. **a** Prostate glands from PB-Cre4, *Pten*^flox/flox^ mice were H&E stained or probed for total PTK6 (DAB, brown), active AKT (PS473, green), active PTK6 (PY342, green), active FAK (PY576/577 and PY925, green), and phospho-BCAR1 (PY165, green). Black size bar, 500 μM; White size bar, 20 μM. **b** Immunohistochemistry of PTK6 (DAB, brown) and active PTK6 PY342, (brown) in different lobes of the mouse prostate gland. VP, ventral prostate; DLP, dorsolateral prostate; AP, anterior prostate. Normal rabbit IgG was used as a negative control. Size bar, 50 μm. **c** Fields of cells staining positive for Ki67 (green) in PB-Cre4, *Pten*^flox/flox^, *Ptk6*^+/+^ and PB-Cre4, *Pten*^flox/flox^, *Ptk6*^−/−^ mice were measured by immunofluorescence and quantified. Size bar, 20 μm (*n* = 6, ***P < 0.001). **d** Fields of cells staining positive for BrdU incorporation (green) in PB-Cre4, *Pten*^flox/flox^, *Ptk6*^+/+^ and PB-Cre4, *Pten*^flox/flox^, *Ptk6*^−/−^ mice were measured by immunofluorescence and quantified. Size bar, 20 μm (*n* = 6, ***P < 0.001). **e** Prostate glands from PB-Cre4, *Pten*^flox/flox^, *Ptk6*^+/+^ were double stained for PTK6 PY342 (red) and Cytokeratin 8 (green, left panel) or Cytokeratin 14 (green, right panel). Colocalization of proteins is observed by an additive fluorescence signal (yellow). Scale bar, 20 μm

G129R[24], the protein phosphatase dead PTEN mutant Y138L[30], and the lipid phosphatase-dead PTEN mutant G129E[20, 31]. Mutation of four clustered serine/threonine residues at the carboxy terminus of PTEN (S380, T382, T383, and S385) to alanine (A4) relieves PTEN inhibition[32–34], and we also tested the activities of the PTEN mutants bearing the A4 mutation alone or in combination with the protein phosphatase Y138L and dual phosphatase-deficient C124S and G129R mutations (Fig. 4b). Membrane targeted Palm-PTK6-WT and Palm-PTK6-YF are both phosphorylated at Y342. Palm-PTK6-WT is also phosphorylated at the inhibitory tyrosine residue 447, which is absent in Palm-PTK6-YF because it has been mutated to phenylalanine. In the presence of PTEN G129E, which lacks lipid phosphatase activity but retains protein phosphatase activity, phosphorylation at Y342 is removed in both Palm-PTK6-WT and Palm-PTK6-YF, while introduction of inactive PTEN C124S and G129R, as well as the protein phosphatase dead PTEN Y138L have no impact. Mutation of the PTEN inhibitory phosphorylation sites within its C-terminal tail (A4) did not enhance the ability of Y138L to dephosphorylate PTK6 Y342, nor did it affect the activity of any of the catalytically inactive PTEN mutants (Fig. 4b). These data indicate that PTEN protein phosphatase activity targets activating phosphorylation of PTK6.

We regularly see 30–40% transfection efficiency in the PC3 cell line. However, in our experiments, transfected PTEN appeared to target PTK6 PY342 in most cells (Fig. 4b). Recently, a few studies have suggested that active PTEN may be secreted and taken up by neighboring cells[35–37]. With high transient expression of PTEN in transfected cells, some functional PTEN may be exported into the supernatant and taken up by the surrounding cells that have not been transfected. It will be interesting to examine this possibility in future studies.

Interestingly, the A4 single mutant, which is defective in autoinhibition, only targeted Palm-PTK6-WT PY342 but not Palm-PTK6-YF PY342. None of the PTEN constructs influenced phosphorylation at the PTK6 autoinhibitory residue Y447 (Fig. 4b). We performed coimmunoprecipitation experiments and found that PTK6 Y447 is required for PTEN A4 association. Wild type PTK6 but not PTK6 YF associated with PTEN A4 in a coimmunoprecipitation experiment (Fig. 4c), explaining the inability of the PTEN A4 mutant to target PTK6 YF.

We performed immunofluorescence studies and the results for cells expressing Palm-PTK6-WT or Palm-PTK6-YF and empty vector, wild type PTEN, PTEN Y138L, PTEN G129E, and PTEN A4 are shown (Fig. 4d). Both wild type and lipid phosphatase defective PTEN G129E target PTK6 Y342 phosphorylation notably in peripheral adhesion complexes, which are most evident when Palm-PTK6-YF is expressed. PTEN with mutation of its protein phosphatase activity (Y138L) activity did not target PTK6 Y342 phosphorylation. Moreover, in agreement with data from Fig. 4a, the PTEN A4 single mutant displays activity toward PTK6 PY342 when PTK6-WT is expressed but not PTK6-YF. Even though it contains the palmitoylation/myristoylation membrane targeting signal, the pool of Palm-PTK6-WT that is phosphorylated on tyrosine residue 447 is not membrane-associated, and none of the different PTEN constructs had an impact on Y447 phosphorylation (Supplementary Fig. 1B).

To examine specificity of PTEN for PTK6 PY342, we compared its ability to target PY342 with a panel of recombinant tyrosine phosphatases, including PTP1B, SHP1, and the catalytic domain of the transmembrane phosphatase CD45. An earlier study demonstrated that PTP1B directly dephosphorylates PTK6 at PY342[38]. Equal amounts of protein tyrosine phosphatases were incubated with PTK6 for varying time intervals, and their abilities to dephosphorylate PTK6 at PY342 and/or PY447 was measured using immunoblotting. Both PTEN and PTP1B dephosphorylated

PTK6 at Y342 in a time dependent manner (Fig. 4e). SHP1 and the cytoplasmic catalytic domain of CD45 had no effect on PY342. None of the phosphatases tested had any effect on PY447 of PTK6. Taken together, these data suggest that PTEN regulates PTK6 activity by dephosphorylating PY342 without affecting inhibitory phosphorylation at Y447.

**PTK6 promotes murine prostate cancer progression following loss of PTEN function.** Prostate-specific disruption of *Pten* in the mouse leads to invasive adenocarcinoma[39]. Probasin (PB)-Cre4, *Pten*[flox/flox] mice display high grade prostatic intraepithelial neoplasia (HGPIN) at 9 weeks of age progressing to invasive carcinoma from 17 to 42 weeks[40]. We previously detected high levels of endogenous active membrane-associated PTK6 (PY-342) in tumors that formed following conditional disruption of *Pten* in this model[14]. Here we have shown that the protein phosphatase activity of PTEN negatively regulates PTK6-PY342 activating phosphorylation. To determine the contributions of PTK6 activation to prostate gland tumorigenesis following loss of *Pten*, we crossed the PB-Cre4, *Pten*[flox/flox] mice with *Ptk6*[−/−] mice. Mouse prostate glands were examined after 8 months which is a sufficient interval for tumors to form in PB-Cre4, *Pten*[flox/flox] mice[39]. At eight months of age, tumors were examined by a veterinary pathologist and subjected to histological grading[41]. *Pten*[flox/flox], and *Pten*[flox/flox], *Ptk6*[−/−] mice, which did not harbor the PB-Cre4 allele, did not display any pathological phenotype. Out of the 11 PB-Cre4, *Pten*[flox/flox] mice examined, 6 developed invasive adenocarcinoma of the prostate gland (Table 1). In contrast, mice lacking *Ptk6* were protected from prostate cancer. Out of the 11 PB-Cre4, *Pten*[flox/flox], *Ptk6*[−/−] mice examined, only one developed adenocarcinoma of the prostate gland ($P < 0.05$). Interestingly, disruption of *Ptk6* also impaired development of urethral carcinomas in the PB-Cre4, *Pten*[flox/flox] mice. More than 25% of PB-Cre4, *Pten*[flox/flox] mice (3/11) developed urethral carcinomas, in contrast with 0/11 PB-Cre4, *Pten*[flox/flox], *Ptk6*[−/−] mice. These results demonstrate a critical role for *Ptk6* in cell transformation following *Pten* loss of function in the prostate.

We examined activation of signaling pathways in the prostates of the eight month old mice described above. We detected activating phosphorylation of FAK on tyrosine residues 576/577[42], as well as phosphorylation of FAK tyrosine residue 925 that mediates its interaction with the adapter protein GRB2[43] in prostates of PB-Cre4, *Pten*[flox/flox] mice. Phosphorylation of the adapter protein BCAR1 on tyrosine residue 165 was also detected following disruption of *Pten* (Fig. 5a). We did not detect tyrosine phosphorylation of either of these PTK6 substrates in PB-Cre4, *Pten*[flox/flox], *Ptk6*[−/−] mice (data not shown). In both genotypes, active Akt (PS473) was easily detected (Fig. 5a), but phospho-STAT3 was not observed.

PTK6 expression was detected in all lobes (ventral, dorsal-lateral, and anterior) of the wild type eight month old mouse prostate, with significant levels in the nuclei of epithelial cells throughout. However, membrane-specific activation of PTK6 was most striking in the anterior lobe (AP) following disruption of *Pten* in mice (Fig. 5b). We also examined PTK6 activation (PY342) in younger mice, aged 8 weeks and 16 weeks (Supplementary Fig. 2). PTK6 activation was most striking at apical membranes in eight week old PB-Cre4, *Pten*[flox/flox], but this polarity decreased by 16 weeks of age, when PTK6 PY342 was also detected basolaterally. Activation of PTK6 at the plasma membrane was not detected in mice expressing wild type *Pten*.

In the tumors collected at eight months of age, cell proliferation was monitored by analyzing Ki67 expression or BrdU incorporation using immunofluorescence. Cells positive for Ki67 or BrdU within tumors of both genotypes were counted and

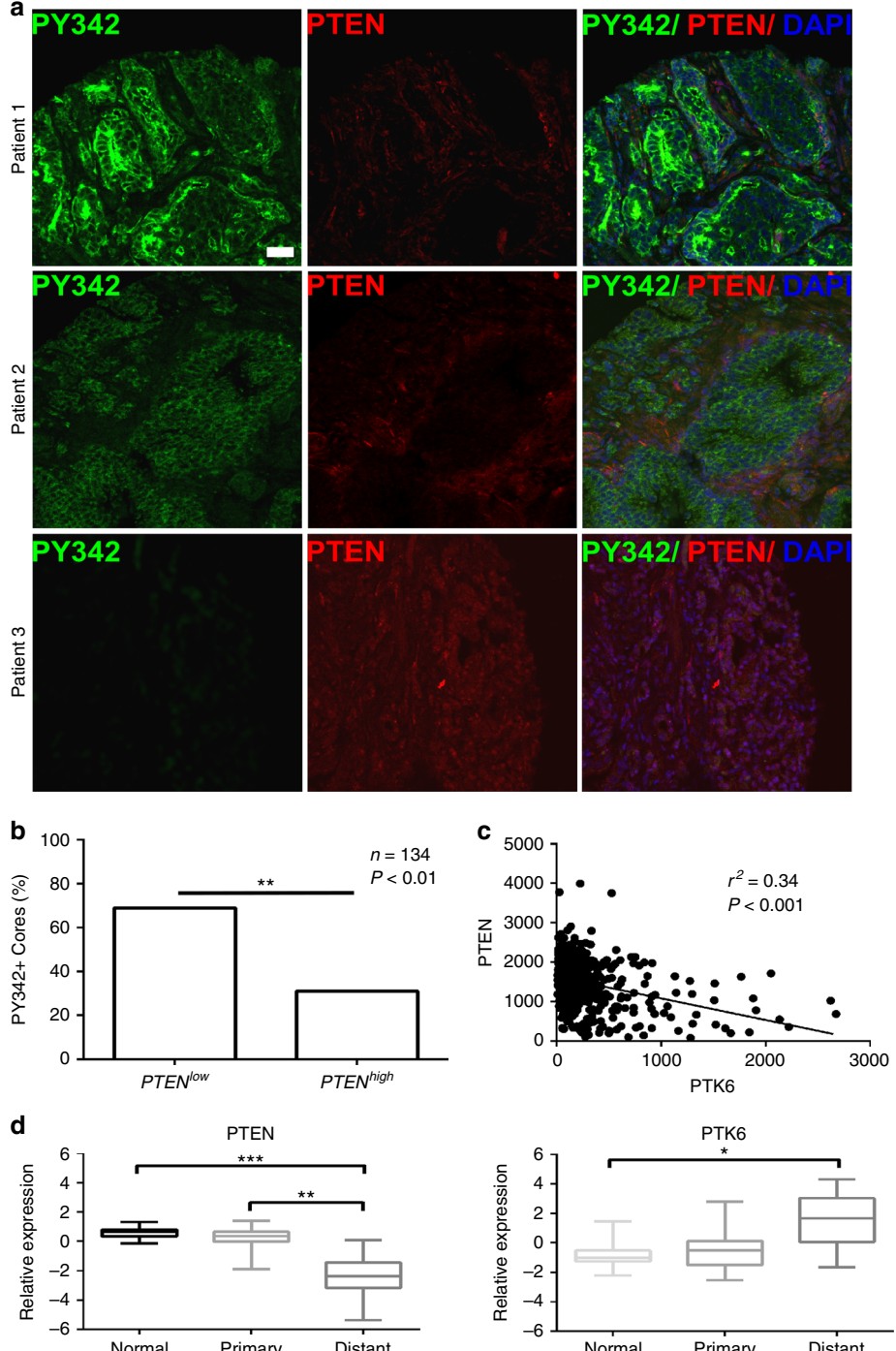

**Fig. 6** PTK6 is activated in human prostate tumors that lack PTEN expression. **a** Immunofluorescence was performed on CPCTR TMA1 using antibodies specific for active PTK6 (PY342, green) and total PTEN (red). Scale bar, 50 μm. **b** Human prostate tumor cores displaying high level of active PTK6 were scored for PTEN expression. ($n = 131$, $^{**}P < 0.01$) **c** Coexpression of PTEN and PTK6 mRNA from The Cancer Genome Atlas (TCGA). Linear regression analysis was performed. **d** PTEN and PTK6 mRNA expression data exported from the Grasso Prostate data set[49] accessed via Oncomine ($^{***}P < 0.001$, $^{**}P < 0.01$, $^{*}P < 0.05$)

quantified. The number of Ki67 + or BrdU incorporating cells was significantly reduced in the PB-Cre4, $Pten^{flox/flox}$, $Ptk6^{-/-}$ mice as compared to PB-Cre4, $Pten^{flox/flox}$, $Ptk6^{+/+}$ mice (Fig. 5c, d). Taken together, these data suggest that activation of FAK and phosphorylation of BCAR1 are dependent on PTK6 in $Pten$ null tumors, and cells lacking PTK6 are less proliferative than PTK6 wild-type cells.

To examine the origin of the cells in which PTK6 was activated in the $Pten$-deficient prostate, we determined if PTK6 PY342 was localized to epithelial cells expressing either the luminal marker cytokeratin 8 (CK8) or the basal cell marker cytokeratin 14 (CK14)[44]. We detected PTK6 PY342 in luminal cells that express CK8, while CK14-positive basal cells were negative for active PTK6 (Fig. 5e).

**Table 1 Effects of *Ptk6* disruption in the PB-Cre4, *Pten* flox/flox mouse model of prostate cancer**

| Genotype | No. animals with PIN/ total | No. animals with invasive adenocarcinoma/ total | No. animals with urethral carcinoma/ total |
|---|---|---|---|
| *Pten* flox/flox | 0/3 (0%) | 0/3 (0%) | 0/3 (0%) |
| *Pten* flox/flox, *Ptk6*−/− | 0/3 (0%) | 0/3 (0%) | 0/3 (0%) |
| PB-Cre4, *Pten* flox/flox | 11/11 (100%) | 6/11* (55%) | 3/11 (27%) |
| PB-Cre4, *Pten* flox/flox, *Ptk6*−/− | 11/11 (100%) | 1/11 (9%) | 0/11 (0%) |

PIN prostatic intraepithelial neoplasia
The table summarizes the pathological phenotypes observed for each genotype[41]
*P < 0.05, unpaired *t*-test

**PTEN loss correlates with activation and increased expression of PTK6 in human prostate cancer**. Since we detected striking activation of PTK6 in a mouse model of prostate cancer, we aimed to better understand the clinical significance of PTK6 activation in prostate cancer patients. We probed TMA1 generated by the Cooperative Prostate Cancer Tissue Resource (CPCTR)[45, 46], which contains cancer tissue from radical prostatectomy specimens from 299 patients, including control non-neoplastic tissue from benign prostatic hyperplasia (BPH) and control non-diseased tissue from organ donor prostates, as well as cores from two prostate cancer cell lines: LNCaP and PC-3. Using immunofluorescence and confocal microscopy, we observed that loss of PTEN expression correlates with activation of PTK6 (Fig. 6a). Conversely, tumors from patients with normal PTEN expression display low levels of active PTK6 (Fig. 6b). These data suggest that activation of PTK6 could be a biomarker for advanced disease, and a subset of patients with both loss of PTEN expression and activation of PTK6 might benefit from a drug regimen including PTK6 inhibitors.

PTEN expression has been identified as a biomarker for prostate cancer patients and inversely correlates with Gleason score and pathologic stage[47]. An inverse correlation between PTK6 and PTEN expression was observed after analyzing gene expression data from The Cancer Genome Atlas (TCGA) Prostate Adenocarcinoma (PRAD) dataset[48]. Using linear regression analysis, we calculated a statistically significant ($P < 0.001$) $r^2$ value of 0.34 (Fig. 6c). We also examined RNA expression from a cohort of prostate cancer patient samples in the Grasso data set available on Oncomine[49], and found that loss of PTEN expression also coincides with an increase in PTK6 messenger RNA (mRNA) (Fig. 6d). These data suggest PTEN contributes to negative regulation of PTK6 gene expression in addition to directly negatively regulating PTK6 activating phosphorylation in the prostate.

**Discussion**
PTEN is an essential tumor suppressor that functions as a lipid phosphatase to negatively regulate phosphatidylinositol 3 kinase/ AKT signaling (reviewed in ref. [17]). It also acts as a dual specificity protein phosphatase[22], but this activity of PTEN is less well-studied and understood. It has been proposed that the protein phosphatase activities of PTEN are important for its auto-dephosphorylation and autoregulation[50]. In addition, several signaling proteins involved in cancer, including Rab7[51], focal adhesion kinase[52, 53], IRS1[54], CREB[55], and the PDGF receptor[56] have been identified as PTEN protein substrates. Here we demonstrate that PTK6 PY342 is a PTEN protein phosphatase substrate. We show that wild type PTEN and a lipid phosphatase-defective PTEN mutant that retains protein phosphatase activity target phosphorylation of PTK6 at Y342. In contrast, the PTEN Y138L mutant that lacks protein phosphatase activity and two

different catalytically inactive PTEN mutants did not target PTK6 Y342. Our data indicate that the protein phosphatase activity of PTEN is specific for the activating phosphorylation of PTK6 at Y342; PTEN does not target phosphorylation at PTK6 Y447. Even PTEN with mutation of the negative regulatory phosphorylation sites in its carboxy terminus was unable to target PTK6 Y447 (Fig. 4).

Historically, phosphatases have been thought to be promiscuous and to function generically to attenuate the kinome (reviewed in ref. [57]). However in recent years, growing evidence for phosphatase specificity has emerged. For example, Tonks and colleagues demonstrated that PTP1B directly and specifically targets PTK6 (BRK) PY342 leading to its inhibition, while it indirectly promotes activation of the structurally related SRC kinase[38, 58]. Here we compared the abilities of the phosphatases PTEN, PTP1B, SHP1, and CD45 to target phosphorylated tyrosine residues in PTK6. Both PTEN and PTP1B targeted activating phosphorylation at Y342, but not inhibitory phosphorylation at Y447 (Fig. 4e). SHP1 and CD45 did not target either residue in PTK6. Our data provide further evidence of that phosphatases display specificity in their recognition of substrates.

Evidence that PTK6 might be regulated by PTEN was first provided by our observations that active membrane-associated PTK6 was present in mouse prostates following conditional deletion of *Pten* and in the *PTEN*−/− PC3 human prostate cancer cell line[14]. Knockdown of PTK6 in PC3 cells inhibited xenograft tumor growth and metastasis, while overexpression of membrane targeted active PTK6 promoted tumorigenesis and the EMT. Areas of the mouse prostate glands with high levels of active membrane-associated PTK6 also had reduced expression of E-cadherin, and further analysis demonstrated that activation of PTK6 is correlated with the EMT in vivo[14]. These studies suggested that active PTK6 might play a significant role in promoting prostate tumorigenesis following loss of PTEN. Here we demonstrate that disruption of *Ptk6* in the PB-Cre4, *Pten*flox/flox model impairs prostate tumorigenesis (Table 1).

Activation of PTK6 at the plasma membrane was associated with increased activating phosphorylation of FAK and phosphorylation of the adapter protein BCAR1 in PB-Cre4, *Pten*flox/flox prostate glands, but we could not detect phosphorylation in PB-Cre4, *Pten*flox/flox, *Ptk6*−/− mice. FAK and BCAR1 interact with and are direct substrates of PTK6[15, 25]. FAK was identified as one of the first protein substrates of PTEN[53], but lack of FAK phosphorylation only in the PB-Cre4, *Pten*flox/flox, *Ptk6*−/− prostates suggests that PTK6 is required for phosphorylation of FAK in vivo. PTK6, FAK, and BCAR1 are all implicated in regulating cell adhesion, placing these proteins in the same signaling networks.

We detected total PTK6 in nuclei of luminal epithelial cells in all lobes of the wild type mouse prostate gland (Fig. 5b), and activation of PTK6 primarily in CK8 positive luminal cells in the anterior prostate following prostate-specific disruption of *Pten*

(Fig. 5b, e). Prostate cancers may originate from either luminal or basal cells[59], and PB-Cre4 is expressed in both cell types[60]. Luminal cells have been shown to be the preferred cells of origin for prostate cancer in multiple mouse models[61], and cancers that originate from luminal cells may have a worse outcome[62].

In agreement with our mouse studies, we found PTK6 activation at the plasma membrane inversely correlated with loss of PTEN expression in clinical samples represented on TMA1 generated by the CPCTR[45]. Interestingly, by examining PTK6 and PTEN gene expression in the Grasso[49] and TCGA[48] prostate cancer datasets, we also found PTK6 mRNA expression is often inversely correlated with PTEN gene expression. In addition, Grasso and colleagues also examined gene copy number alterations in 61 prostate cancer patients, and gain of PTK6 copy number correlated with loss of PTEN in 13/14 samples. Cumulatively, these data indicate that PTK6 and PTEN are coregulated at multiple levels in prostate cancer. In addition to loss of PTK6 nuclear localization and activation at the plasma membrane, increased expression also may contribute to the development of prostate cancer.

While loss of PTEN leads to increased PI3K/AKT signaling, targeting the PI3K/AKT signaling pathway alone utilizing PI3K inhibitors or rapamycin analogs has had suboptimal therapeutic efficacy. A variety of data suggest that PTEN substrates outside of the canonical PI3K/AKT/mTOR axis are clinically important. Our data indicate that combination therapies incorporating inhibition of PTK6 signaling may have added benefits in the treatment of PTEN-deficient prostate cancers.

## Methods

**Antibodies**. Antibodies specific for indicated proteins were obtained from the following sources: Santa Cruz Biotechnology (Dallas, TX): human PTK6 (Brk C-18, sc-1188, 1:1,000), mouse PTK6 (Brk C-17, sc-916, 1:100 for IHC) and monoclonal PTK6 (Brk G-6, sc-166171, 1:1,000); FAK (C-20, sc-558, 1:1,000 for IB, 1:100 for IF), E-cadherin (sc-7870, 1:1,000), phosphotyrosine (PY20, sc-508, 1:5,000) Cytokeratin 14 (sc-53253, 1:100 for IF), and SP1 (PEP2, sc-59, 1:1,000); Millipore (Billerica, MA): EGFR (06-847, 1:1,000), phosphotyrosine (4G10, 05-321, 1:5,000) and PY342-PTK6 (09-144, 1:1,000 for IB, 1:100 for IF); Abcam (Cambridge, MA): PY447-PTK6 (ab138368, 1:1,000 for IB, 1:100 for IF), Ki67 (ab16667, 1:100 for IF) and Cytokeratin-8 (ab59400, 1:100 for IF); Cell Signaling Technology (Danvers, MA): AKT (9272, 1:1,000 for IB, 1:100 for IF), P-AKT (Thr-308, 9275, 1:1,000), P-AKT (Ser-473, 9271, 1:1,000 for IB), P-AKT (Ser-473, IHC, 9277, 1:100), P-BCAR1 (Tyr-165, 4015, 1:1,000 for IB, 1:100 for IF), P-FAK (Tyr-576/577, 3281, 1:1,000 for IB, 1:100 for IF), P-FAK (Tyr-925, 3284, 1:1,000 for IB, 1:100 for IF), HA-tag (6E2, 2367, 1:1,000), PTEN (138G6, 9559, 1:1,000) and Myc tag (2276, 1:1,000); BD Biosciences (San Jose, CA): BCAR1 (610271, 1:1,000 for IB, 1:100 for IF); Vector Laboratories (Burlingame, CA): normal rabbit IgG (I-1000, 1:5,000); Sigma-Aldrich (St. Louis, MO): β-actin (AC-15, A5441, 1:10,000) and vinculin (V9264, 1:10,000); Cascade Bioscience (Winchester, MA) PTEN (6H2.1, ABM-2052, 1:100); Covance (Cumberland, VA): GST tag (4C10, MMS-112R, 1:1,000); Donkey anti-rabbit (NA934, 1:500) or sheep anti-mouse (NA931, 1:500) antibodies conjugated to horseradish peroxidase were purchased from GE Healthcare (Pittsburgh, PA) and detected by chemiluminescence with SuperSignal West Dura substrate (34075) from Thermo Fisher Scientific (Waltham, MA).

**Expression constructs**. Full-length human PTK6, PTK6-KM (kinase dead), and PTK6-YF (active) containing an N-terminal MYC epitope tag in the pcDNA3 vector with and without addition of a palmitoylation/myristoylation signal have been described[29]. GST-tagged human PTK6 was cloned into pGEX-2T using the BamHI and EcoRI restriction sites. GST-tagged mouse PTK6 fusion proteins (Full-length, SH1, SH2, SH3, SH2-SH3) in pGEX-KG have been described[63]. HA-tagged PTEN and mutant PTEN G129R in the pBABE-puro vector have been described[24]. Constructs encoding HA-tagged WT PTEN, PTEN C124S, PTEN G129R, PTEN A4, PTEN G129E, PTEN C124S A4, and PTEN G129R A4 in the pSG5L vector were acquired from Addgene Inc. (Cambridge, MA). 401 pSG5L (Addgene plasmid #10737), 800 pSG5L HA PTEN WT (Addgene plasmid #10750), 811 pSG5L HA PTEN C124S (Addgene plasmid #10744), 817 pSG5L HA PTEN G129R (Addgene plasmid #10748), 977 pSG5L HA PTEN A4 (Addgene plasmid #10753), 882 pSG5L HA PTEN G129E (Addgene plasmid #10746), 1276 pSG5L HA PTEN C124S A4 (Addgene plasmid #10949), and 1241 pSG5L HA PTEN G129R (Addgene plasmid #10948) were gifts from William Sellers. GST-tagged full-length human PTEN in the bacterial expression vector pGEX-3X (Addgene plasmid #20743) was a gift

from Alonzo Ross[64] and acquired from Addgene. The pMKO.1-puro-vector (Addgene plasmid #8452) was a gift from Bob Weinberg[65] The pMKO.1-puro-PTEN shRNA (Addgene plasmid #10669) construct was a gift from William Hahn and acquired from Addgene. Prostate cancer cell lines PC3 (ATCC CRL-1435) and DU145 (ATCC HT-81) were cultured according to ATCC guidelines. Transfections were performed using Lipofectamine 2000 purchased from Invitrogen (Carlsbad, CA) according to manufacturer's instructions. pBABE-puro or pMKO.1-puro vectors were used to generate stable pools of cell lines that were selected in media containing 2 μg/ml puromycin, as previously described[26].

**Site-directed mutagenesis**. HA-tagged PTEN Y138L and HA-tagged PTEN Y138L A4 constructs were created using the Q5 Site-Directed Mutagenesis kit from New England Bio Labs (Ipswich, MA). HA-tagged PTEN Y138L and HA-tagged PTEN Y138L A4 inserts in the pSG5L vector were used as the backbone. For both constructs, the tyrosine encoded by the TAT codon at residue 138 was changed to leucine using the CTG codon. The reaction was carried out using the forward primer 5′-GATATGTGCACTGTTATTACATCGGGGCAAATTTTTAAAG-3′ and reverse primer 5′-ATTACACCAGTTCGTCCC-3′ following the manufacturer's instructions. The TAT to CTG mutation is underlined.

**Protein lysates, cell fractionation, and immunoprecipitation**. Cells with ectopic PTEN expression or PTEN knockdown were harvested only at early passages. Total cell lysates were prepared as previously described[26]. Cell fractionation was carried out using the ProteoExtract subcellular proteome extraction kit (Calbiochem, Billerica, MA) according to manufacturer's instructions. Immunoprecipitations were carried out, as previously described[26].

**Immunostaining**. Cells cultured on 8-well chamber slides were fixed in Carnoy's solution (ethanol-chloroform-acetic acid, 6:3:1). For mouse tissues, prostate glands were formalin-fixed and paraffin-embedded. Tissues were subjected to immunostaining as described[14]. For immunofluorescence, proteins were detected using goat anti-mouse Alexa Fluor 488, goat anti-rabbit Alexa Fluor 488, streptavidin-conjugated Alexa Fluor 488 or streptavidin-conjugated Alexa Fluor 594 purchased from Invitrogen (Carlsbad, CA). For immunohistochemistry, proteins were detected using horseradish peroxidase-conjugated secondary antibodies in the presence of 3,3′-diaminobenzidine (DAB). Tissues for immunofluorescence were counterstained with 4′,6-diamidino-2-phenylindole (DAPI).

**GST fusion protein purification and pull-down assay**. The GST-tagged PTK6 fusions and PTEN proteins were expressed in BL21 Star (DE3)pLysS cells (Life Technologies, Carlsbad, CA). Protein expression was induced with the addition of 1 mM isopropyl β-D-1-thiogalactopyranoside (IPTG) and purification was carried out as previously described[63]. For GST pull-down experiments, 5 μg of murine PTK6 GST-fusion proteins (GST, SH3, SH2, SH3/SH2, SH1, and full-length) were incubated with 30 μl glutathione-Sepharose 4B beads (50% slurry) for 30 min; 500 of μg of DU145 total cell lysate was then added and incubated overnight at 4 °C. Beads were washed 4 times in wash buffer (1% Triton X-100, 20 mM HEPES, pH 7.4, 150 mM NaCl, 1 mM EDTA, 1 mM EGTA, 10 mM Na-pyrophosphate); resuspended in 2× SDS loading buffer then boiled for 5 min and resolved by SDS–PAGE.

**In vitro phosphatase and kinase assays**. The in vitro phosphatase assay was performed according to Shi et al.[54] with slight modifications. Human GST-tagged PTK6 protein was incubated with an increasing concentration of human GST-tagged PTEN protein in phosphatase buffer (20 mM HEPES, pH 7.5, 50 mM NaCl, 5 mM MgCl2 1 mM DTT, and 0.1% NP-40) at 30 °C for 1 h. The reaction was stopped with the addition of 6× SDS loading buffer and samples were subjected to SDS–PAGE. Purified PTEN, PTP1B, SHP1, and the intracellular domain of CD45 were obtained from Enzo Life Sciences (Farmingdale, NY, USA). In total 8 ng of purified phosphatase was added to 8 ng of GST-tagged PTK6 in 50 μl of phosphatase buffer for indicated times. The reaction was stopped with the addition of 6× SDS loading buffer and samples were subjected to SDS–PAGE.

For in vitro kinase assays, Myc-tagged human PTK6-YF was transfected into PC3 vector, PC3 PTEN, and PC3 G129R stable cell lines. Protein lysates were subjected to immunoprecipitation[26] using anti-Myc tag antibody and 50% slurry A/G Agarose (Santa Cruz Biotechnology). Washed beads were resuspended with kinase reacting solution (Takara Clontech, Mountain View, CA). The in vitro kinase assay was performed using the Universal Tyrosine Kinase Assay Kit (Takara Clontech, USA) according to the manufacturer's instructions.

**Murine prostate cancer models**. Generation and characterization of PB-Cre4 and Pten^flox/flox mice have been described[39]. We crossed C57BL/6J PB-Cre4 Pten^flox/flox mice with C57BL/6 Ptk6^−/− mice to generate triple transgenic PB-Cre4 Pten^flox/flox Ptk6^−/− mice. The genetic model was confirmed by PCR analysis of the PB-Cre4 allele, Pten^flox/flox allele, and deletion of Ptk6 exon 1. Mice were sacrificed at 8 months They were injected with 5-bromo-2-deoxyuridine (BrdU) (Sigma) in phosphate-buffered saline at 50 μg/g of body weight 1 h prior to being sacrificed. Specimens were scored blindly using a labeling system known only to the group

providing the specimens. Experiments were reviewed and approved by the University of Illinois at Chicago Institutional Animal Care and Use Committee.

**Human tissue microarray analysis.** Prostate cancer tissue microarray (TMA) slides were obtained from the National Cancer Institute (NCI) Cooperative Prostate Cancer Tissue Resource (CPCTR). CPCTR TMA1 contains 299 patient samples presented in quadruplicate[45]. Slides were dual-stained for active PTK6 (PY342) and PTEN as described above. In total 134 core biopsies from 105 individual patients displaying high levels of active PTK6 (PY342) were scored for PTEN status independently by two researchers.

**Statistics.** Immunoblot quantification was determined by ImageJ[66]. For all experiments, at least 3 independent trials were performed. Results are shown as the mean + SE. For mRNA expression analysis, data from the Grasso Prostate data set[49] was obtained via Oncomine. The data was comprised of 122 samples—35 castrate-resistant metastatic prostate cancers, 59 localized prostate adenocarcinomas, and 28 benign prostate tissues. For analysis of immunoblot intensities and gene expression, P-values were determined using a two-tailed, unpaired Student's $t$ test (GraphPad Prism 6) and observed variation within and across groups was similar. For mouse studies, sample sizes were chosen to allow detection of the group average difference in tumor number that is about the same magnitude of within-group standard deviation at approximately an 80% power and 0.05 significance level. No animals were excluded from scoring. Gene expression data from The Cancer Genome Atlas (TCGA) Prostate Adenocarcinoma (PRAD) data set was accessed via cBioPortal. Linear regression analysis was utilized to examine coexpression of PTK6 and PTEN. For analysis of CPCTR TMA1, the quantity of PY342$^{high}$PTEN$^{low}$ and PY342$^{high}$PTEN$^{high}$ cores was subjected to a one-tailed binomial distribution test. A difference was considered statistically significant if the P-value was less than 0.05.

**Data availability.** The authors declare that all data are available within the manuscript or supplementary files, or available from the authors upon request.

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

## Acknowledgements

These studies were supported by NIH grant R01CA188427 (A.L.T.). We thank Dr Vadim Gaponenko and Milica Gilic for helpful discussion of the work.

## Author contributions

D.J.W.: Acquisition of data, analysis and interpretation of data, drafting the manuscript. A.-K.B.: Acquisition of data and analysis and interpretation of data. V.M.: Acquisition of data and analysis and interpretation of data. S.B.-K.: Analysis and interpretation of data. M.L.Z.: Analysis and interpretation of data. W.B.: Acquisition of data and analysis and interpretation of data. A.L.T.: Study concept and design, analysis and interpretation of data, drafting the manuscript.

## Additional information

**Competing interests:** The authors declare no competing financial interests.

