## [Peer Review File · Nature Communications]

Reviewers' comments:

Reviewer #1 (Remarks to the Author):

Wozniak et al have investigated a link between the tumor suppressor phosphatase PTEN and the tyrosine kinase PTK6/BRK/SIK. Their data show in multiple prostate cancer cell types that PTEN can inhibit the phosphorylation of the activating tyrosine residue in PTK6, Tyr342, and the phosphorylation of downstream PTK6 substrates such as BCAR1/p130CAS. They also show that PTK6 and PTEN appear to physically interact and that PTK6 is required for rapid prostate tumorigenesis in mouse prostates lacking PTEN. I find most of the data interesting, novel and convincing and appreciate the rigor with which the study has been conducted. However, I think there is insufficient data to conclude that the major claim that the dephosphorylation of PTK6 induced by PTEN is direct.

An expansion of the functional link between PTEN and PTK6 alone may be important enough to merit publication, although this would appear much stronger if it was investigated in several other cell types and lineages.

Evidence for direct dephosphorylation: Expression of PTEN and PTEN G129E in PTEN null cell lines such as PC3 has been shown to induce major changes in cellular processes such as proliferation (see below mostly WT PTEN only) and migration (observed with PTEN G129E) which are associated with changes in the phosphorylation of many hundreds of protein phosphorylation sites. Almost all of the observed phosphorylation changes in this type of experiment should be expected to be indirect and therefore a high burden of proof should be required to claim an unlikely direct relationship. With rather weak experimental approaches currently available, demonstrating direct dephosphorylation of any substrate by a dedicated protein phosphatase is an extremely challenging task. If the authors wish strongly to claim a direct enzyme-substrate relationship, they should:

1. Test the in vitro dephosphorylation of PTK6 by PTEN alongside a few of the 80 or so dedicated human protein tyrosine phosphatases and showing the purity of the purified PTEN proteins (protein stain not western blot).
2. Present a full set of data with quantitative comparison of the dephosphorylation of PTK6 by PTEN wt, G129E and inactive mutants. Current data show G129E vs C124S and separately WT and G129R.
3. It's notable that PTEN WT binds to PTK6 but G129R does not. To exclude artefacts caused by over-expression of PTEN G129E and PTEN which has been reported to bind to numerous signalling proteins, they should test the effect on PTK6 phosphorylation of PTEN mutants which selectively lack protein phosphatase activity yet have an open/active conformation via mutation of C-terminal inhibitory phosphorylation sites (362/366/370 and 380/382/383/385). It would be good to use the Y138L lipid phosphatase active mutant within these mutants, e.g. Y138L 366A or Y138L 380/382/383/385A mutants.

Minor Points:

The use of stable clones. The authors describe the generation of stable clones expressing PTEN. Data investigating the effect of PTEN expression are simply described as PTEN stably expressing without clarification whether the data are generated using a single individual (representative?) clone, a freshly pooled oligoclonal mixture etc. This should be made clear. If data are presented for one clone, it should be stated how many individual clones have been tested (and perhaps whether any variability in response was observed, whether this correlated with PTEN expression level etc).

It is worth noting that this referee is very cautious of data generated with stable clones of tumor cells engineered to re-express tumor suppressor genes which they have lost during their oncogenic development. There are many reports of the induction of growth arrest in PTEN null cells including PC3 upon its re-expression (e.g. Van Duijn, Prostate, 2010; Persad et al, PNAS, 2000). In most

cases it seems unlikely that such cells will proliferate substantially in culture without unknown additional mutations/changes and the validity of such an approach should be demonstrated (eg describe what frequency of transfected cells gave rise to a proliferative puromycin resistant clone for empty vector and for PTEN). The approach assumes that other observed changes in clones, eg PTK6 phosphorylation, are caused by PTEN and not (many days or perhaps weeks later) by other unknown differences between the experimental cell clone sets.

Introduction: The incidence figures look either like they may be only for the USA. This should be specified.

Does the sustained pY signal in the 4th panel of Fig 4A imply that there are many other autophosphorylated tyrosine residues on PTK6 which are not substrates for PTEN? This is an interesting point and perhaps a useful control.

Reviewer #2 (Remarks to the Author):

The study investigates the relationship of PTEN and PTK6 in prostate cancer using human cell line studies in vitro and mouse models in vivo. The authors also provide clinical data the last figure.

The main conclusion is that PTEN phosphorylates PTK6, which is interesting but it is hard to see the broad relevance of this work for the readers of nature communication. It would seem that this study may be more appropriate for a more specialized journal.

Major concerns:

Studies are limited to using human prostate cancer cell lines that are lacking and/or have impaired androgen receptor signaling (PC3 and DU145), which is a major issue considering the importance of PTEN for AR signaling in prostate cancer.

If it is a germline mutation of Ptk6 (Ptk6^{-/-}) this reviewer thinks it is difficult to say with certainty that the phenotypic consequences are cell autonomous or even restricted to or dependent solely on changes that occur in the prostate.

Some of the data is not very rigorous or fully/adequately described. Some examples:

Figure 1 - there are no markers to confirm fractionation of the cells (so for example, a marker expressed in membranes or nuclear fraction). Also the data in Panel B are not very striking and it is difficult to see how the quantification matches to the western.

Figure 2 - the knock-down is quite subtle, as is the effects on the effector proteins.

Figure 5 - the phenotype of the mouse models are not fully described - and need to be to make the conclusions stronger.

Other Specific comments:

References - the authors has some unusual choices and also seem to be lacking key references. (for example - #2, 12, 27)

Results - para 1 - this reviewer does not understand the following sentence (lacking context)

"Our observation that Pten loss in the mouse prostate led to activation of PTK6 at the plasma membrane led us to hypothesize that PTEN may directly negatively regulate PTK6 in

human prostate cancer cells."

In the results - para 2 (and corresponding figure) - this reviewer find the use of the abbreviations to be confusing (and unnecessary) - e.g. "We expressed MYC-tagged wild-type (WT), kinase-dead (K219M, denoted KM) or inhibition-defective (Y447F, denoted YF)"

The point below is a critical one that should be confirmed in the cell lines by sequencing - the immunoblot only confirms that the protein is expressed not that it is wild-type.

"The prostate cancer cell line DU145 expresses wild-type PTEN (Figure 2A)."

In the comment below, how was it confirmed that the bacterially-expressed protein was folded properly and active?

"Recombinant GST-tagged PTK6 and PTEN proteins were overexpressed and purified from E. coli."

Note that the time frame for the emergence of tumors in the current mice is different than that published in ref 26.

Reviewer #3 (Remarks to the Author):

Nature Communications-16-19168

Title: PTEN is a Protein Phosphatase for PTK6 and Inhibits PTK6 Oncogenic Signaling in Prostate Cancer

Recommendation: Rejection

In this manuscript, Darren et al. report that PTEN can act as a protein phosphatase towards PTK6, which in turn, regulates oncogenic pathway in prostate cancer progression. Biochemical assays show that both WT and lipid phosphatase-inactive PTEN mutants, retaining the intact protein phosphatase activity, could target active PTK6 at Y342 residue in the plasma membrane compartment both in vivo and in vitro. The authors go on to show that genetic ablation of Ptk6 in the Pb-Cre, Ptenflox/flox mice, suppresses Pten KO-mediated prostate tumor progression and they also explore the clinical relevance of this PTEN/PTK6 pathway in the human cancer specimens. Identifying relevant PTEN protein phosphatase substrates and the mechanisms underlying metastatic cancer progression would be certainly of great interest. However, the data presented in this manuscript are preliminary and, in several instances, do not support the authors' conclusions and rather contradictory. What is even more concerning is that the authors did not rigorously compare PTEN protein phosphatase-inactive Y138L mutant with PTEN WT, G129E (Lipid phosphatase inactive) or C124S (both lipid and protein phosphatase inactive) mutants towards PTK6 phosphorylation simultaneously in any experiments throughout the manuscript. Moreover, the study lacks novelty in general, since the authors have already reported that active pY342 PTK6 accumulated in the plasma membrane of PTEN null prostate cancer cells and mice prostates following conditional deletion of Pten, which in turn contributed to EMT and metastasis in xenografted tumor model (PMID: 23856248). Nevertheless, the specific points included below should be addressed to improve the quality of the data and support the conclusions of this study.

Major points:

1) In Fig. 1A-D, the authors should also include PTEN Y138L and C124S mutants as a control to further corroborate that PTK6 is a novel protein phosphatase substrate of PTEN.

2) In Fig. 2, the authors claim that PTK6 can act as a protein phosphatase of PTEN by shRNA-mediated PTEN silencing approach. However, this data does not support the role of PTK6 as a

bona-fide protein phosphatase substrate of PTEN, since it is well known that knockdown of PTEN also hyper-activates PI3K-AKT signaling pathway. The authors should reintroduce either WT, G129E, C124S, or Y138L PTEN variants into Pten KO cells to further clarify the functions of PTEN towards PTK6.

3) In Fig.3B, Co-Immunoprecipitation assays revealed that PTEN preferentially interacts with WT and a kinase-deficient PTK6, but not with active YF PTK6. However, in Fig. 1C-D, overexpression of PTEN still could efficiently remove phospho-Y342 residue from constitutively active YF PTK6, which is inconsistent with the Co-IP data. As such, the authors' need to reconcile this difference in relation to the data that they are showing.

4) Similar to the previous critique, in Fig. 3C, PTK6 seems to preferentially interact with WT PTEN, but not G129R PTEN mutant, which is, once again, in contradiction with the data shown in Fig. 1A demonstrating that overexpression of G129R mutant could behave like its WT counterpart to dephosphorylate PTK6 at Y342 residue. The authors should reconcile this difference in relation to the data that they are showing.

5) In Fig. 6A-B, loss of PTEN seems to be associated with the increase of PTK6 expression in human clinical specimens. However, the data presented here does not support the working hypothesis that PTK6 is a novel protein phosphatase substrate of PTEN at all. Instead, it directly contradicts other data throughout the manuscript, showing that overexpression of PTEN does not affect PTK6 level in cancer cell lines. If that were to be the case, it is highly likely that the increase of PTK6 phosphorylation mediated by PTEN loss is caused by the elevation of total PTK6 expression. Along this line, quantification of PTK6 phosphorylation by normalization with its total protein level is needed in Fig. 6C-D.

Minor points:

1) In Fig. 5B, the IHC image showing the decrease of Ki67 mediated by Ptk6 loss should be included.

2) In Fig. 5D, authors should include the images with separate green and red colors in order to be able to more easily to distinguish that pY342 PTK6 colocalized with luminal cell marker Cytokeratin 8 (CK8), but not basal cell marker Cytokeratin 14 (CK14).

Response to Reviewers:

We thank the reviewers for their very helpful comments and suggestions. In response to the review, we have added several additional experiments; we have reorganized the figures and have added two data panels to Figures 1 and 2, one to Figure 3, and three to Figure 5. A supplementary figure is also now included. The data contained in revised Figure 4 are new and address major concerns of reviewers 1 and 3. In addition to wild type PTEN, we include results from several mutant PTEN constructs including dual phosphatase-defective (C124S, G129R), protein phosphatase-dead (Y138L), lipid phosphatase-dead (G129E), and inhibition relieved (A4; S380 T382 T383 S385 to A) alone, or in combination with other mutations (Y138L,A4; C124S,A4; G129R,A4). Experiments with all of these constructs were performed at the same time and these new data are included in Figure 4. Our responses to the previous review are provided in bold font below.

Reviewer #1:

If the authors wish strongly to claim a direct enzyme-substrate relationship, they should:

1. Test the in vitro dephosphorylation of PTK6 by PTEN alongside a few of the 80 or so dedicated human protein tyrosine phosphatases and showing the purity of the purified PTEN proteins (protein stain not western blot).

We have included the Coomassie stain of the protein blot as requested in Figure 3B. It would not be surprising to find that other phosphatases regulate PTK6. In fact, Tonks and colleagues reported that the protein tyrosine phosphatase PTB1 also targets PTK6 phosphorylation (Fan, et al., 2015). Although there is a pervading view that phosphatases lack specificity, PTB1 specifically inhibited PTK6 activation while it promoted SRC activation. Here we present data that PTEN protein phosphatase activity is highly specific for PY342 and does not impact PY447, providing an internal control for PTEN specificity for phosphorylated tyrosines within the same protein.

2. Present a full set of data with quantitative comparison of the dephosphorylation of PTK6 by PTEN wt, G129E and inactive mutants. Current data show G129E vs C124S and separately WT and G129R. 3. It's notable that PTEN WT binds to PTK6 but G129R does not. To exclude artefacts caused by over-expression of PTEN G129E and PTEN which has been reported to bind to numerous signalling proteins, they should test the effect on PTK6 phosphorylation of PTEN mutants which selectively lack protein phosphatase activity yet have an open/active conformation via mutation of C-terminal inhibitory phosphorylation sites (362/366/370 and 380/382/383/385). It would be good to use the Y138L lipid phosphatase active mutant within these mutants, e.g. Y138L 366A or Y138L 380/382/383/385A mutants.

3. It's notable that PTEN WT binds to PTK6 but G129R does not. To exclude artefacts caused by over-expression of PTEN G129E and PTEN which has been reported to bind to numerous signalling proteins, they should test the effect on PTK6 phosphorylation of PTEN mutants which selectively lack protein phosphatase activity yet have an open/active conformation via mutation of C-terminal inhibitory phosphorylation sites (362/366/370 and 380/382/383/385). It would be good to use the Y138L lipid phosphatase active mutant within these mutants, e.g. Y138L 366A or Y138L 380/382/383/385A mutants.

As suggested, we now include data from a full panel of mutants in the same figure, including the PTEN A4 mutant alone, and in combination with the protein phosphatase and catalytic PTEN mutants. Figure 4 is new and shows that the protein phosphatase activity of PTEN specifically targets the activating phosphorylation of PTK6 (PY342) and not inhibitory phosphorylation (PY447).

Our new data indicate that the carboxy tails of PTK6 and PTEN are involved in mediating an interaction between the two proteins, as we now show that the PTEN A4 mutant associates with wild type PTK6, but not PTK6 YF (Fig. 4C). Understanding the mechanisms of PTEN - PTK6 interaction will be an interesting focus of future studies, but are beyond the focus of the current study.

Minor Points:

We utilized stable cell pools shortly after selection, only at low passages. PC3 cells stably expressing Vector, WT PTEN and PTEN G129R were selected in puromycin for one week. After one week, PTEN protein was observed by western blot along with changes in PTK6 phosphorylation. In accordance with the reviewer and previous publications, PC3 cells stably expressing PTEN fail to grow in our lab after 7-10 passages. When this occurred, cells frozen directly after selection were plated, PTEN expression was confirmed by western blot, and cells were used for future experiments.

We thank the reviewer for noting the possibility of other genetic alterations affecting PTK6 phosphorylation. Our data, as shown in Figure 4B and 4D, suggest that PTEN alone is sufficient to reduce activation of PTK6. Cells in these experiments were not stably expressing PTEN; rather, they were transiently transfected with PTEN and PTEN recombinant mutants and harvested after 24 hours. This time frame would likely not allow for significant alterations. After only 24 hours, WT PTEN, PTEN G129E, and PTEN A4 were able to dephosphorylate PTK6 PY342 without affecting phosphorylation of PY447. Phosphatase-dead PTEN C124S, phosphatase-deficient PTEN G129R and protein phosphatase-dead Y138L had no effect on PTK6 phosphorylation at residue 342 or 447 further affirming that the protein phosphatase activity of PTEN is directly responsible for this effect.

The incidence figures cited in the introduction are for the United States and this is now indicated.

Does the sustained pY signal in the 4th panel of Fig 4A imply that there are many other autophosphorylated tyrosine residues on PTK6 which are not substrates for PTEN? This is an interesting point and perhaps a useful control.

Both Y342 (activating) and Y447 (inhibiting) are sites for PTK6 autophosphorylation. We show that PY342 but not PY447 is a PTEN substrate in the new Figure 4.

Reviewer #2:

The main conclusion is that PTEN *phosphorylates* PTK6, which is interesting but it is hard to see the broad relevance of this work for the readers of nature communication. It would seem that this study may be more appropriate for a more specialized journal.

We show that PTEN *dephosphorylates* PTK6 PY342, which is important for its activation, thereby inhibiting its oncogenic signaling activities in the prostate. Loss of PTEN leads to PTK6 activation and promotes its oncogenic membrane-associated signaling. We believe this is of broad relevance, because it helps to explain how the loss of PTEN promotes prostate cancer.

Major concerns:

Studies are limited to using human prostate cancer cell lines that are lacking and/or have impaired androgen receptor signaling (PC3 and DU145), which is a major issue considering the importance of PTEN for AR signaling in prostate cancer.

Here we have focused on PTEN and PTK6 and utilize PTEN negative and positive cell lines, as well as the in vivo mouse model. We agree that exploring the relationship between AR and PTK6 will be interesting for future studies, but it is not relevant to the current work.

If it is a germline mutation of Ptk6 (*Ptk6*^{-/-}) this reviewer thinks it is difficult to say with certainty that the phenotypic consequences are cell autonomous or even restricted to or dependent solely on changes that occur in the prostate.

The *Ptk6*^{-/-} mouse model has been extensively characterized and we have included a new paragraph describing this mouse model in more detail, including several additional references in the introduction. No phenotype has been detected in the *Ptk6*^{-/-} prostate. However, here we show that loss of *Ptk6* in the *Pten* null prostate impairs prostate tumorigenesis.

The included cell line and in vitro studies support a cell autonomous mechanism.

Some of the data is not very rigorous or fully/adequately described. Some examples:

Figure 1 - there are no markers to confirm fractionation of the cells (so for example, a marker expressed in membranes or nuclear fraction). Also the data in Panel B are not very striking and it is difficult to see how the quantification matches to the western.

We routinely probe cell fractions for membrane, cytoplasmic and nuclear markers, but do not always show them due to space limitations. We have included all of these controls in Figure 2.

Figure 2 - the knock-down is quite subtle, as is the effects on the effector proteins.

The knockdown is sufficient to induce activation of AKT and PTK6.

Figure 5 - the phenotype of the mouse models are not fully described - and need to be to make the conclusions stronger.

We have described the mouse model in more detail in the introduction and we have added Figure 5C and 5D showing differences in Ki67 expression and BrdU incorporation in the mouse model.

Other Specific comments:

References - the authors has some unusual choices and also seem to be lacking key references. (for example - #2, 12, 27)

The references cited by the reviewer include recent reviews in Cancer Discovery (2), Annual Reviews in Biochemistry (12) and a review by Robert Matusik on the mouse model (27) that the authors feel are relevant. We apologize if we have omitted some other key references due to space limitations.

Results - para 1 - this reviewer does not understand the following sentence (lacking context)

"Our observation that Pten loss in the mouse prostate led to activation of PTK6 at the plasma membrane led us to hypothesize that PTEN may directly negatively regulate PTK6 in human prostate cancer cells."

In the results - para 2 (and corresponding figure) - this reviewer find the use of the abbreviations to be confusing (and unnecessary) - e.g. "We expressed MYC-tagged wild-type (WT), kinase-dead (K219M, denoted KM) or inhibition-defective (Y447F, denoted YF)"

We have revised the sentence that was causing confusion. We find it difficult not to use the abbreviations, particularly in figure labeling. WT, KM, YF are standard notations for the mutations and they have been defined in the text.

The point below is a critical one that should be confirmed in the cell lines by sequencing - the immunoblot only confirms that the protein is expressed not that it is wild-type.

"The prostate cancer cell line DU145 expresses wild-type PTEN (Figure 2A)."

We have changed wild type to functional.

In the comment below, how was it confirmed that the bacterially-expressed protein was folded properly and active?

"Recombinant GST-tagged PTK6 and PTEN proteins were overexpressed and purified from E. coli."

We show that the bacterially expressed PTK6 and PTEN are active in the in vitro experiments shown in Figure 3. We have also now included a Coomassie stained gel showing the purity of the proteins (Figure 3B).

Note that the time frame for the emergence of tumors in the current mice is different than that published in ref 26.

It is well recognized that animals raised in different colonies and having different genetic backgrounds may exhibit some differences in phenotypes. The model described in ref. 26 in 2003 may have been exposed to different environmental factors/colony pathogens, and was not crossed with our *Ptk6* model. We agree with the reviewer that in our lab, *Pten*^{flox/flox} PB-Cre4 mice develop tumors later than originally described in Ref. 26. Our mice were sacrificed at 8 months (32 weeks) and mice developed cancer with 55% penetrance. This is in contrast to Reference 26 (median age of cancer, 4 months (16 weeks). Yet, several subsequent studies have shown that this model develops invasive adenocarcinoma at varying ages and with differing penetrance. For example, Chen *et al* (2005) reports a 4-6 month latency (16-24 weeks) and Zhang *et al* (2009) described that invasive carcinoma developed within a range of 20-40 weeks.

Reviewer #3:

In this manuscript, Darren et al. report that PTEN can act as a protein phosphatase towards PTK6, which in turn, regulates oncogenic pathway in prostate cancer progression. Biochemical assays show that both WT and lipid phosphatase-inactive PTEN mutants, retaining the intact protein phosphatase activity, could target active PTK6 at Y342 residue in the plasma membrane compartment both in vivo and in vitro. The authors go on to show that genetic ablation of *Ptk6* in the Pb-Cre, *Pten*^{flox/flox} mice, suppresses *Pten* KO-mediated prostate tumor progression and they also explore the clinical relevance of this PTEN/PTK6 pathway in the human cancer specimens. Identifying relevant PTEN protein phosphatase substrates and the mechanisms underlying metastatic cancer progression would be certainly of great interest. However, the data presented in this manuscript are preliminary and, in several instances, do not support the authors' conclusions and rather contradictory. What is even more concerning is that the authors did not rigorously compare PTEN protein phosphatase-inactive Y138L mutant with PTEN WT, G129E (Lipid phosphatase inactive) or C124S (both lipid and protein phosphatase inactive) mutants towards PTK6 phosphorylation simultaneously in any experiments throughout the manuscript. Moreover, the study lacks novelty in general, since the authors have already reported that active pY342 PTK6 accumulated in the plasma membrane of PTEN null prostate cancer cells and mice prostates following conditional deletion of *Pten*, which in turn contributed to EMT and metastasis in xenografted tumor model (PMID: 23856248). Nevertheless, the specific points included below should be addressed to improve the quality of the data and support the conclusions of

this study.

It should be noted that while our previous study included the observation that high levels of active PTK6 could be detected at the plasma membrane in some cells in the *Pten* null prostate, these data were descriptive and only provided the rationale for the current study. The previous paper did not address PTEN regulation of PTK6 activity and the significance of PTK6 activation in the *Pten* null mouse prostate and in human prostate tumor tissue. The data presented in this manuscript are novel.

Major points:

1) In Fig. 1A-D, the authors should also include PTEN Y138L and C124S mutants as a control to further corroborate that PTK6 is a novel protein phosphatase substrate of PTEN.

2) In Fig. 2, the authors claim that PTK6 can act as a protein phosphatase of PTEN by shRNA-mediated PTEN silencing approach. However, this data does not support the role of PTK6 as a bona-fide protein phosphatase substrate of PTEN, since it is well known that knockdown of PTEN also hyper-activates PI3K-AKT signaling pathway. The authors should reintroduce either WT, G129E, C124S, or Y138L PTEN variants into *Pten* KO cells to further clarify the functions of PTEN towards PTK6.

We claim that PTEN is a protein phosphatase for PTK6. Knockdown of PTEN leads to increased activating phosphorylation of PTK6 on a tyrosine residue, as well as increased activation of AKT, a serine/threonine kinase. As suggested, we have included a full panel of mutant PTEN constructs in the new Figure 4 including protein phosphatase dead PTEN Y138L, and demonstrate that the protein phosphatase activity is critical for the dephosphorylation of PTK6 by PTEN. The lipid phosphatase-dead PTEN mutant G129E still effectively targets PTK6 (Fig. 4B), while it would not have an impact on AKT activity.

3) In Fig.3B, Co-Immunoprecipitation assays revealed that PTEN preferentially interacts with WT and a kinase-deficient PTK6, but not with active YF PTK6. However, in Fig. 1C-D, overexpression of PTEN still could efficiently remove phospho-Y342 residue from constitutively active YF PTK6, which is inconsistent with the Co-IP data. As such, the authors' need to reconcile this difference in relation to the data that they are showing.

4) Similar to the previous critique, in Fig. 3C, PTK6 seems to preferentially interact with WT PTEN, but not G129R PTEN mutant, which is, once again, in contradiction with the data shown in Fig. 1A demonstrating that overexpression of G129R mutant could behave like its WT counterpart to dephosphorylate PTK6 at Y342 residue. The authors should reconcile this difference in relation to the data that they are showing.

PTEN interacts with PTK6 Y447F sufficiently to reduce PY342 signal. This binding may be slightly weaker than that of PTEN to PTK6 WT or PTK6 K219M, as shown by Co-IP in Figure 1E. Yet, the binding between PTEN and PTK6 Y447F is not ablated. New data indicate that the PTEN A4 mutant cannot interact with PTK6 YF, but interacts with and dephosphorylates wild type PTK6. These studies underscore a role for the PTK6 carboxy-terminal tail in mediating interactions with PTEN that merits future investigation.

As shown in Figure 1A, 1B, and Supplementary Figure 1, PTEN G129R does not function similarly to WT PTEN. The phosphatase activity of PTEN is required to reduce activation of PTK6 as PTEN G129R has no effect on PTK6 PY342 signal. The Co-IP data further suggest that PTEN G129R does not act on PTK6 as these data suggest PTEN G129R does not bind PTK6. Rather, PTEN must be wild-type to bind PTK6 and possess phosphatase activity to dephosphorylate PTK6

5) In Fig. 6A-B, loss of PTEN seems to be associated with the increase of PTK6 expression in human clinical specimens. However, the data presented here does not support the working hypothesis that PTK6 is a novel protein phosphatase substrate of PTEN at all. Instead, it directly contradicts other data throughout the manuscript, showing that overexpression of PTEN does not affect PTK6 level in cancer cell lines. If that were to be the case, it is highly likely that the increase of PTK6 phosphorylation mediated by PTEN loss is caused by the elevation of total PTK6 expression. Along this line, quantification of PTK6 phosphorylation by normalization with its total protein level is needed in Fig. 6C-D.

We have reorganized Figure 6 to emphasize that loss of PTEN expression correlates with PTK6 activation at the plasma membrane in patient samples (Figure 6 A and B). These data correlate well with our mouse data, and suggest that active PTEN is required for PTK6 inhibition in human prostate. Additional data that PTK6 mRNA is upregulated in prostate tumors that have lost PTEN are presented in Fig. 6 C - E, and suggest additional modes of regulation. These data are not contradictory. A modest increase in PTK6 expression cannot account for the dramatic activation of PTK6 at the plasma membrane. Earlier studies of PTK6 protein expression did not show a significant increase in protein expression in high grade/stage prostate tumors (Derry et al., 2003).

Minor points:

1) In Fig. 5B, the IHC image showing the decrease of Ki67 mediated by Ptk6 loss should be included.

We now include the Ki67 expression, as well as BrdU incorporation data in new Figure 5 C and D.

2) In Fig. 5D, authors should include the images with separate green and red colors in order to be able to more easily to distinguish that pY342 PTK6 colocalized with luminal cell marker Cytokeratin 8 (CK8), but not basal cell marker Cytokeratin 14 (CK14).

Individual channels are now shown in Figure 5E.

Reviewers' comments:

Reviewer #1 (Remarks to the Author):

I think some of the new data is very strong and adds to the manuscript, but major issues have not all been resolved. I would argue strongly against making bold claims about a direct versus indirect mechanism when available experimental methods to address this question are weak. I think much of the data is very good and aspects of the work are worthy of publication without an answer to this point.

The authors have not addressed the first point raised in review of their first submission. They need to compare the activity of PTEN to dephosphorylate PTK6 in vitro against other tyrosine phosphatases (e.g. purified PTP1B, SHP1, CD45, PTPN12 are all available from multiple sources) and calculate reaction rates. PTEN is an evolutionarily conserved lipid phosphatase and its protein phosphatase activity is weak against the (best) substrates against which it has been most thoroughly analysed. If PTK6 is a real direct PTEN protein substrate, it is reasonable to expect that PTEN should be able to dephosphorylate PTK6 in vitro with good kinetics. Showing a 1hr time point, analysed by blotting with no other phosphatases as controls is not convincing.

Fig 4B. This is potentially an important piece of data. PC3 Cells which have been stably transfected with PTK6 are transiently transfected with PTEN expression vectors. How is PTK6 phosphorylation reduced below detectable levels in a cell lysate by PTEN phosphatase activity if only a subset of the cells are transfected? Are transfection rates (% of cells) approaching 100%? This seems unlikely. E.g. even transfection reagents optimised by the manufacturers to be specific for the PC3 line are only claimed to achieve 50% transfection efficiency in marketing literature.

The method for the in vitro dephosphorylation experiments should be clarified as it is not obvious how bacterially expressed PTK6 is phosphorylated – autophosphorylation? incubation with mammalian cell lysate?

Reviewer #3 (Remarks to the Author):

Nature Communications-NCOMMS-16-19168A-Z

In the revised version of the manuscript, the authors have provided additional data that improve the overall quality of the study and also provide further mechanistic insight to support the proposed model. Therefore, a more compelling case for PTEN to directly dephosphorylate PTK6 and in turn regulates oncogenic pathway in prostate cancer progression is presented in the text. However, there are still some remaining concerns that should be addressed prior to publication. First, the authors should mention the age of the mice throughout the manuscript, as some of this information is currently missing. Second, using western blotting and immunohistochemistry approaches to further corroborate the role of PTK6 as a bona-fide substrate of PTEN, the authors should show the significant increase of pY342 PTK6 levels in young Pb-Cre4, Ptenflox/flox mice (e.g. two months old).

Response to Referees

We thank the reviewers for their insightful comments, which are addressed point by point below.

Reviewer #1

"They need to compare the activity of PTEN to dephosphorylate PTK6 in vitro against other tyrosine phosphatases (e.g. purified PTP1B, SHP1, CD45, PTPN12 are all available from multiple sources) and calculate reaction rates. PTEN is an evolutionarily conserved lipid phosphatase and its protein phosphatase activity is weak against the (best) substrates against which it has been most thoroughly analysed. If PTK6 is a real direct PTEN protein substrate, it is reasonable to expect that PTEN should be able to dephosphorylate PTK6 in vitro with good kinetics. Showing a 1 hr time point, analysed by blotting with no other phosphatases as controls is not convincing."

As suggested, we purchased active PTEN, PTP1B, SHP1, and the intracellular catalytic domain of CD45 from Enzo Life Sciences and compared the abilities of these purified phosphatases to dephosphorylate PTK6 over time. PTP1B provides a positive control because Tonks and colleagues have shown that PTP1B directly targets PY342 in PTK6 (BRK) ^{1, 2}. In new Figure 4E, we show that PTEN is as efficient as PTP1B in dephosphorylating PTK6 PY342. Neither SHP1 nor CD45 targeted PTK6 in these assays. Equivalent amounts of purified phosphatase and substrate were used in each reaction.

"Fig 4B. This is potentially an important piece of data. PC3 Cells which have been stably transfected with PTK6 are transiently transfected with PTEN expression vectors. How is PTK6 phosphorylation reduced below detectable levels in a cell lysate by PTEN phosphatase activity if only a subset of the cells are transfected? Are transfection rates (% of cells) approaching 100%? This seems unlikely. E.g. even transfection reagents optimised by the manufacturers to be specific for the PC3 line are only claimed to achieve 50% transfection efficiency in marketing literature."

Using GFP-tagged expression constructs, we regularly see 30 - 40% transfection efficiency using Lipofecamine 2000 in the PC3 cell line. Recently, a few studies have suggested that active PTEN may be secreted and taken up by neighboring cells ^{3, 4, 5}. It is possible that with high transient expression of PTEN in transfected cells, some functional PTEN is exported into the supernatant and taken up by the surrounding cells that have not been transfected. However, while it will be interesting to test this possibility, it is beyond the scope of the current study.

"The method for the in vitro dephosphorylation experiments should be clarified as it is not obvious how bacterially expressed PTK6 is phosphorylated – autophosphorylation? incubation with mammalian cell lysate?"

Bacterially expressed PTK6 is capable of autophosphorylation at Y342 ⁶ and Y447 ⁷.

Reviewer #3

"First, the authors should mention the age of the mice throughout the manuscript, as some of this information is currently missing."

We now mention the age of the mice used throughout the results section.

Second, using western blotting and immunohistochemistry approaches to further corroborate the role of PTK6 as a bonafide substrate of PTEN, the authors should show the significant increase of pY342 PTK6 levels in young PbCre4, Ptenflox/flox mice (e.g. two months old)."

As Reviewer 3 suggested, we have done immunohistochemistry for active PTK6 with prostates from 8 and 16 week old mice and these data are presented in new supplementary Figure 2. Active PTK6 is detected following disruption of *Pten* in the prostate in younger mice at these earlier time points, but not in age-matched control mice expressing *Pten*.

REFERENCES

1. Fan G, Lin G, Lucito R, Tonks NK. Protein-tyrosine phosphatase 1B antagonized signaling by insulin-like growth factor-1 receptor and kinase BRK/PTK6 in ovarian cancer cells. *The Journal of Biological Chemistry* **288**, 24923-24934 (2013).
2. Fan G, Aleem S, Yang M, Miller WT, Tonks NK. Protein-tyrosine Phosphatase and Kinase Specificity in Regulation of SRC and Breast Tumor Kinase. *J Biol Chem* **290**, 15934-15947 (2015).
3. Putz U, Mah S, Goh CP, Low LH, Howitt J, Tan SS. PTEN secretion in exosomes. *Methods* **77-78**, 157-163 (2015).
4. Putz U, et al. The tumor suppressor PTEN is exported in exosomes and has phosphatase activity in recipient cells. *Sci Signal* **5**, ra70 (2012).
5. Hopkins BD, et al. A secreted PTEN phosphatase that enters cells to alter signaling and survival. *Science* **341**, 399-402 (2013).
6. Qiu H, Miller WT. Regulation of the nonreceptor tyrosine kinase Brk by autophosphorylation and by autoinhibition. *J Biol Chem* **277**, 34634-34641 (2002).
7. Takeda H, et al. Comparative analysis of human SRC-family kinase substrate specificity in vitro. *J Proteome Res* **9**, 5982-5993 (2010).

REVIEWERS' COMMENTS:

Reviewer #1 (Remarks to the Author):

The authors have dealt well with the issues raised after the initial revision.

Most importantly, I think the new data 4E looks impressive and convincing

The authors should add short clarifying notes in the results text relating to Fig 4B explaining that transfection efficiency was assessed as 30-40% and that suppression in untransfected cells requires an alternate mechanism (perhaps intercellular PTEN transfer as they propose). Similarly that bacterially expressed PTK6 can autophosphorylate (that might only need a single well placed word).

Reviewer #3 (Remarks to the Author):

In their revised version of the manuscript, "PTEN is a Protein Phosphatase that Targets Active PTK6 and Inhibits PTK6 Oncogenic Signaling in Prostate Cancer", the authors have provided additional data that collectively improve the quality of the study and also provide further mechanistic insight to support the proposed model.

As such, the authors now provide a more compelling case for PTK6 as a bona-fide protein phosphatase substrate of PTEN, which contributes to prostate cancer progression. On this basis, the authors additionally offer a potential therapeutic modality for treating PTEN deficient prostate cancers by targeting PTK6 signaling. Since our previous concerns have been addressed, I believe that the manuscript is now ready for publication in Nature Communications.

REVIEWERS' COMMENTS:

Reviewer #1 (Remarks to the Author):

The authors have dealt well with the issues raised after the initial revision.

Most importantly, I think the new data 4E looks impressive and convincing.

The authors should add short clarifying notes in the results text relating to Fig 4B explaining that transfection efficiency was assessed as 30-40% and that suppression in untransfected cells requires an alternate mechanism (perhaps intercellular PTEN transfer as they propose). Similarly that bacterially expressed PTK6 can autophosphorylate (that might only need a single well placed word).

Reviewer #3 (Remarks to the Author):

In their revised version of the manuscript, "PTEN is a Protein Phosphatase that Targets Active PTK6 and Inhibits PTK6 Oncogenic Signaling in Prostate Cancer", the authors have provided additional data that collectively improve the quality of the study and also provide further mechanistic insight to support the proposed model. As such, the authors now provide a more compelling case for PTK6 as a bona-fide protein phosphatase substrate of PTEN, which contributes to prostate cancer progression. On this basis, the authors additionally offer a potential therapeutic modality for treating PTEN deficient prostate cancers by targeting PTK6 signaling. Since our previous concerns have been addressed, I believe that the manuscript is now ready for publication in Nature Communications.

Point by point response:

We are pleased that we have been able to address the reviewers' concerns.

As Reviewer 1 suggested, we have added a paragraph discussing transfection efficiency and alternate mechanisms (page 12 of the uploaded manuscript, second paragraph). We have also stated that PTK6 is able to autophosphorylate itself at Y342 and Y447 on page 10, second paragraph.